# Regional tropical cyclone impact functions for globally consistent risk assessments

Samuel Eberenz[1,2], Samuel Lüthi[1,2], David N. Bresch[1,2]

[1] Institute for Environmental Decisions, ETH Zurich, Zurich, 8092, Switzerland

[2] Federal Office of Meteorology and Climatology MeteoSwiss, Zurich-Airport, 8058, Switzerland

*Correspondence to*: Samuel Eberenz (eberenz@posteo.eu)

**Abstract**

Assessing the adverse impacts caused by tropical cyclones has become increasingly important, as both climate change and human coastal development increase the damage potential. In order to assess tropical cyclone risk, direct economic damage is frequently modelled based on hazard intensity, asset exposure and vulnerability, the latter represented by impact functions. In this study, we show that assessing tropical cyclone risk on a global level with one single impact function calibrated for the USA – which is a typical approach in many recent studies – is problematic, biasing the simulated damages by as much as a factor of 36 in the North West Pacific. Thus, tropical cyclone risk assessments should always consider regional differences in vulnerability, too. This study proposes a calibrated model to adequately assess tropical cyclone risk in different regions by fitting regional impact functions based on reported damage data. Applying regional calibrated impact functions within the risk modelling framework CLIMADA at a resolution of 10 km worldwide, we find global annual average direct damage caused by tropical cyclones to range from 51 up to 121 billion USD (current value of 2014, 1980-2017), with the largest uncertainties in the West Pacific basin, where the calibration results are the least robust. To better understand the challenges in the West Pacific and to complement the global perspective of this study, we explore uncertainties and limitations entailed in the modelling setup for the case of the Philippines. While using wind as a proxy for tropical cyclone hazard proves to be a valid approach in general, the case of the Philippines reveals limitations of the model and calibration due to the lack of an explicit representation of sub-perils such as storm surge, torrential rainfall, and landslides. The globally consistent methodology and calibrated regional impact functions are available online as a Python package, ready for application in practical contexts like physical risk disclosure and providing more credible information for climate adaptation studies.

## 1 Introduction

Tropical cyclones (TCs) are highly destructive natural hazards affecting millions of people each year (Geiger et al., 2018; Guha-Sapir, 2018) and causing annual average direct damages in the order of 29 to 89 US$ billions (Cardona et al., 2014; Gettelman et al., 2017; Guha-Sapir, 2018). Climate change and coastal development could significantly increase the impact of TCs in the future (Gettelman et al., 2017; Mendelsohn et al., 2012). Increasing risks from TCs and other extreme weather events pose a challenge to exposed population and assets, but also to governments and investors as actors in globally connected economies. Governments, companies, and investors increasingly express the need to understand their physical risk under current and future climatic conditions (Bloomberg et al., 2017). Thus, quantitative risk assessments require a globally consistent representation of the economic impact of TCs and other natural hazards.

Probabilistic risk models can provide the quantitative basis for risk assessments and adaptation studies. Since the mid-2000s, there have been increasing scientific efforts in developing and improving global scale natural hazard risk assessments

(Cardona et al., 2014; Gettelman et al., 2017; Ward et al., 2020). Risk from natural hazards is frequently modelled as a function of severity and occurrence frequency, which can be computed by combining information on hazard, exposure, and vulnerability (IPCC, 2014). Global and regional scale TC risk models often represent hazard as the spatial distribution of the maximum sustained surface wind speed per TC event (Aznar-Siguan and Bresch, 2019; Ward et al., 2020). In past studies, wind fields modelled from historical TC tracks were used to assess economic risk in the Global Assessment Report (GAR) 2013 (Cardona et al., 2014; UNDRR, 2013) and to quantify affected population (Geiger et al., 2018), among others. For the assessment of future risk, historical TC records can be complemented with events simulated in downscaling experiments based on the output of global climate models (Gettelman et al., 2017; Korty et al., 2017), or synthetic resampling algorithms (Bloemendaal et al., 2020). The exposure component can be represented by the spatial distribution of people, assets or economic values potentially affected by TCs (Geiger et al., 2018; Ward et al., 2020). For the modelling of direct economic damage, exposure is usually derived from building inventories for local risk assessments (Sealy and Strobl, 2017), or estimated by spatially disaggregating national asset value estimates (De Bono and Mora, 2014; Eberenz et al., 2020; Gettelman et al., 2017).

The vulnerability of an exposed value to a given hazard can be represented by impact functions, also called damage functions or vulnerability curves, relating hazard intensity to impact. Impact functions for the assessment of direct economic damage caused by TCs usually relate wind speed to relative damage (Emanuel, 2011). For the USA, TC impact functions are available specific to different building types (Federal Emergency Management Authority [FEMA], 2010; Yamin et al., 2014), as well as on an aggregate level (Emanuel, 2011). Emanuel (2012) found a lack of sensitivity of simulated TC damage to the exact shape of the impact function for the USA. However, due to global heterogeneities in the tropical cyclone climatology (Schreck et al., 2014), building codes, and other socioeconomic vulnerability factors (Yamin et al., 2014), it is inadequate to use a single universal impact function for global TC risk assessments. Bakkensen et al. (2018b) used reported damage data to calibrate TC impact functions for China, highlighting both the potential of this approach and the considerable uncertainties related to the quality of reported damage data. Still, there is a lack of globally consistent and regionally calibrated impact functions. Due to this lack, impact functions calibrated for the USA have been used in a variety of local and regional studies outside the USA, i.e. the Caribbean (Aznar-Siguan and Bresch, 2019; Bertinelli et al., 2016; Ishizawa et al., 2019; Sealy and Strobl, 2017), China (Elliott et al., 2015), and the Philippines (Strobl, 2019). A similar impact function has also been applied for modelling TC damages on a global level (Gettelman et al., 2017). For the GAR 2013, building type specific impact functions from FEMA were assigned to exposure points based on global data based on development level, complexity of urban areas, and regional hazard level at each location (De Bono and Mora, 2014; Yamin et al., 2014). However, the impact functions were not calibrated regionally against reported damage data. Furthermore, the required complexity in exposure data exceeds the scope of many risk assessments.

Can globally consistent TC impact modelling be improved by calibrating the vulnerability component on a regional level? This article addresses this question by calibrating regional TC impact functions in a globally consistent TC impact modelling framework, as implemented within the open-source weather and climate risk assessment platform CLIMADA (Aznar-Siguan and Bresch, 2019). This study contributes to reaching the goal of consistent global TC risk modelling and a better connection of global and regional impact studies. The objectives of this study are to (1) calibrate a global TC impact model by regionalizing the impact function; (2) assess the annual average damage per region and compare the results to past studies; and (3) evaluate the robustness of the calibration and discuss the limitations and uncertainties of both the model setup and the calibration. To inform the discussion of uncertainties, we complement aggregated calibration results (Sect. 3) with an event level case study for the Philippines (Sect. 4). While the attribution of vulnerability to regional drivers is outside the

80 scope of this study, the results can serve as a starting point for further research disentangling the socio-economic and physical drivers determining vulnerability to TC impacts locally and across the globe.

## 2 Data and Method

To regionally calibrate TC impact functions, simulated damages are compared to reported damages, as illustrated in Figure 1: In a first step, direct economic damage caused by TCs are simulated in the impact modelling framework CLIMADA (Fig.

85 1a-d, Sect. 2.1 to 2.2.2) with one single default impact function applied globally to start from (Sect. 2.2.3). Then, damage data points per country and storm are assigned to entries of reported damage (Fig. 1e-f, Sect. 2.3.1). For the matched events, the ratio between simulated and reported damage is calculated (Fig. 1g, Sect. 2.3.2). For calibration, countries are clustered into regions and two complementary cost functions are optimized based on the damage ratios, by regionally fitting the slope of the impact function (Fig. 1h, Sect. 2.3.3).

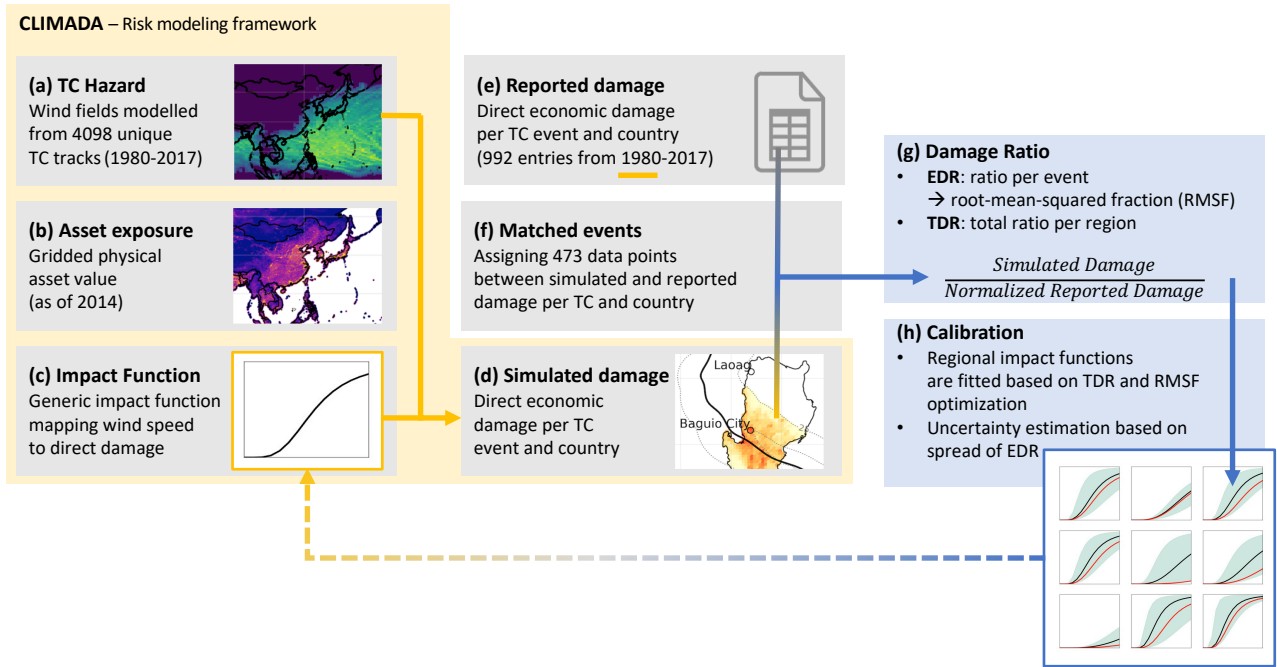

90

**Figure 1: Schematic overview of the data and methods applied to calibrate regional TC impact functions in a globally consistent manner. From left to right: TC event damages are first simulated within the CLIMADA framework based on TC hazard (a), asset exposure (b), and a default impact function (c), c.f. Sect. 2.1 to 2.2.3. Resulting simulated damages (d) are compared to reported damage data from EM-DAT (e) for 473 matched TC events (f) by means of the damage ratio (g), c.f. Sect.**
95 **2.2.4 to 2.3.2. During calibration (h), steps (c) to (g) are repeated several times with varied impact functions for each region, optimizing the cost functions TDR and RMSF (c.f. Sect. 2.3.3) . The result is a set of best fitting impact functions for nine world regions (Sect. 3.2). Finally, the calibrated impact functions are plugged into CLIMADA once more (dashed arrow)to compute annual average damage per region (Sect. 3.3).**

## 2.1 CLIMADA – spatially explicit TC risk modelling
100

The CLIMADA (CLIMate ADAptation) impact modelling framework has been developed at ETH Zurich as a free, open-source software package (Aznar-Siguan and Bresch, 2019). It is written in Python 3.7 and made available online both on GitHub (Bresch et al., 2019a) and the ETH Data Archive (Bresch et al., 2019b). Here, CLIMADA was used for the pre-processing of hazard and exposure data, and for the spatially explicit computation of direct damage on a global grid at 10 km
105 resolution. The setup works equally well at higher chosen resolution, but given uncertainties especially in calibration data and computational constraints justify the chose resolution. In the CLIMADA framework, damage is defined as the product of exposed assets and a damage ratio. The damage ratio is an impact function multiplied with hazard intensity.

In our case, damage per TC event and country is simulated as following: For each grid cell and event, damage is calculated as the product of total exposed asset values and the mean damage ratio. The mean damage ratio (0 to 100%) results from plugging the hazard intensity (maximum sustained wind speed) into the impact function. Finally, damage per event is aggregated over all grid cells within the country. Please refer to Sect. 2.1 and 2.2.3 in Aznar-Siguan and Bresch (2019) for a more detailed description of impact calculation.

## 2.2 Data

### 2.2.1 TC Hazard

TCs typically inflict damage due to strong sustained surface winds, storm surge inundation, and torrential rain (Bakkensen et al., 2018a; Baradaranshoraka et al., 2017; Park et al., 2013). Next to maximum wind speed, storm size is an important factor controlling TC impacts (Czajkowski and Done, 2013). Since the severity of surge and rain are to a certain extend correlated to wind speed and storm size (Czajkowski and Done, 2013), the latter are often taken as a proxy hazard intensity (Emanuel, 2011; Gettelman et al., 2017).

Here, TC hazard intensity is represented by wind fields, i.e. the geographical distribution of the 1-min sustained wind speed per TC event, referred to as "wind speed" or "hazard intensity" in the following. Wind speed was simulated at a horizontal resolution of 10 x 10 km from historical TC tracks as a function of time, location, radius of maximum winds, and central and environmental pressure, based on the revised hurricane pressure-wind model by Holland (2008). Please also refer to Geiger et al. (2018) for a detailed description and illustration of the wind field model and its limitations.

Historical TC tracks were obtained from the International Best Track Archive for Climate Stewardship (IBTrACS) (Knapp et al., 2010). As data quality and global coverage improved after approximately 1980 (Geiger et al., 2018), 4'098 historical TC tracks from 1980 to 2017 was selected based on data completeness criteria with regards to data fields provided within IBTrACS, following the approach described by Geiger et al. (2018) and Aznar-Siguan and Bresch (2019). Out of the 4'098 TCs, a total number of 1'538 landfalling events with the potential of causing damage were identified. Potential damage is given if at least one grid cell of a TCs wind field with an intensity of 25.7 ms$^{-1}$ (~50 knots) or more coincides with an asset exposure value larger than zero. A world map showing the maximum intensity per grid cell for all tracks is shown in the Supplement (Fig. S1).

### 2.2.2 Asset exposure

Asset exposure for the assessment of direct economic risk is represented by the spatially explicit monetary value potentially impacted by a hazard. Here, we use gridded asset exposure value at a resolution of 10 km x 10 km. The dataset is based on the disaggregation of national estimates of total asset value (TAV, Table A3) proportional to the product of nightlight intensity and population count (Eberenz et al., 2020). Following the approach in GAR 2013 (De Bono and Mora, 2014), the TAV per country is represented by produced capital stock of 2014 from the World Bank Wealth Accounting (World Bank, 2019a). Out of the 62 countries used for calibration, 32 come with produced capital estimates. For the remaining 30, an estimate of non-financial wealth is used as a fall back (Eberenz et al., 2020), based on GDP of 2014 from the World Bank Open Data portal (World Bank, 2019b) combined with an GDP-to-wealth factor from the Global Wealth Report (Credit Suisse Research Institute, 2017). The asset exposure dataset utilized here and a detailed overview over limitations and data availability per country is documented in Eberenz et al. (2020).

### 2.2.3 Impact Function

In CLIMADA, vulnerability is represented by impact functions. They are used to compute damage for each TC event at each exposed location by relating hazard intensity to relative impact. Since no directly wind induced damage is expected for low wind speeds, TC impact functions for the spatial explicit modelling of direct damages can be constrained by a minimum threshold $V_{thresh}$ for the occurrence of impacts and an upper bound of a 100% direct damage (Emanuel, 2011). Empirical studies suggest a high power-law function for the slope, i.e. the increase of damage with wind speed (Pielke, 2007). An idealized sigmoidal impact function satisfying these constraints was proposed by Emanuel (2011):

$$f = \frac{v_n^3}{1 + v_n^3} \quad , \text{with} \qquad v_n = \frac{MAX[(V - V_{thresh}), 0]}{V_{half} - V_{thresh}} \qquad \text{(Equation 1)}$$

Equation 1 defines the impact function $f$ as a function of wind speed $V$. The function takes two shape parameters as inputs: $V_{thresh}$ and $V_{half}$. A lower threshold $V_{thresh}$ of 25.7 ms$^{-1}$ (50 kn) was proposed for the USA by Emanuel (2011) and empirically supported for China (Elliott et al., 2015). The slope parameter $V_{half}$ signifies the wind speed at which the function's slope is the steepest and a damage ratio of 50% is reached (Fig. 2). It should be noted that the effects of varying $V_{thresh}$ and $V_{half}$ on resulting impacts are not linearly independent.

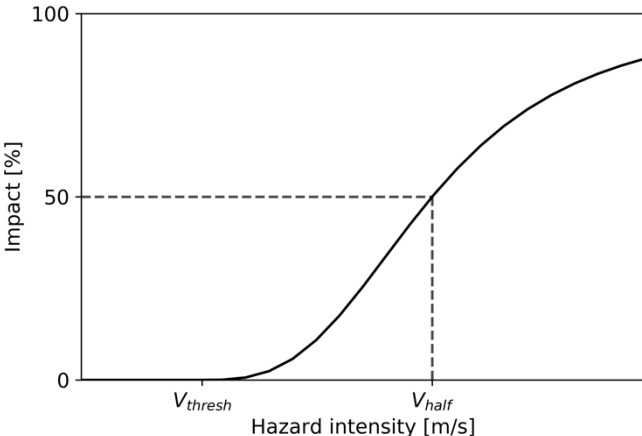

**Figure 2: Idealized TC impact function based on Emanuel (2011). $V_{half}$ is the hazard intensity (i.e. maximum sustained wind speed) at which the relative impact reaches 50% of the exposed asset value. No impact occurs for an intensity below $V_{thresh}$.**

Based on the reference data provided by FEMA (2010), $V_{half}$ for damage to buildings can range from 52 to 89 ms$^{-1}$ depending on building type and surface roughness (Elliott et al., 2015). Applying FEMA impact functions that were verified with reported damage data for US Hurricanes Andrew [1992], Eric [1995], and Fran [1996], Sealy and Strobl (2017) estimated $V_{half}$ to range from 71.7 to 77.8 ms$^{-1}$, depending on building type, with a mean value of 74.7 ms$^{-1}$.

In a comparison of calibration results based on a sigmoidal impact function with a more complex 12-step staircase function, Lüthi (2019) found no improvement of calibration skill with the more complex function. Therefore, a sigmoidal function is applied in this study. The default impact function with $V_{thresh}$ = 25.7 ms$^{-1}$ and $V_{half}$ = 74.7 ms$^{-1}$ is used for a first, uncalibrated, simulation of global TC damages, and as a starting point for calibration. While $V_{half}$ is fitted during the calibration process, the lower threshold $V_{thresh}$ is kept constant throughout the study. This is based on the finding by Lüthi (2019) that the variation of more than one of the linearly dependent parameters most likely results in an overfitting during calibration, with physically implausible values for $V_{thresh}$ in some world regions.

On the chosen 10 km by 10 km grid, single buildings are not resolved. Therefore, damage is aggregated over several buildings in a grid cell and not all buildings are expected to be damaged to the same degree. However, the wind-speed dependent impact function is also implicitly accounting for the damage caused by storm surge and torrential rain, when calibrated against reported damage data. For these two reasons, we allow for values of $V_{half}$ lower and larger than the literature range for pure wind induced building damage in the calibration. To find the functional slope best fit to simulate the direct economic damage of TCs a region, $V_{half}$ is varied step-wise with $V_{half} > V_{thresh}$ (c.f. Sect. 2.3.3, Calibration of regional impact functions).

### 2.2.4 Reported damage data

Reported damage data for historical TC events are required on a global level to calibrate TC impact functions. Reported damage estimates for disasters worldwide are available from the International Disaster Database EM-DAT (Guha-Sapir, 2018). EM-DAT provides data per event and country, including disaster type and subtype, date of the event, and impact estimates. The main data sources of EM-DAT are UN agencies, governmental and non-governmental agencies, reinsurance companies, research institutes, and the press.

EM-DAT provides one entry per country and event. Therefore, one meteorological TC can be listed in EM-DAT several times, with one entry for each country affected. In the following, each of these entries per storm and country will be referred to as single 'TC events'. For instance, Hurricane Irma comes with 17 events in EM-DAT (Disaster no. 2017-0381), as it impacted 16 Caribbean countries and the USA. From 1980 to 2017, there are 1650 TC events reported in EM-DAT of which 991 come with a reported monetary damage value.

The EM-DAT database provides total damage per event and country in current USD. In contrast, the asset exposure data used for the modelling of damage is kept fixed at a current USD value of 2014 (Sect. 2.3). To allow for a comparison of reported and simulated damages that is independent of economic development, reported damage values need to be normalized to a reference year. For instance, Weinkle et al. (2018) applied two normalization methodologies for hurricane damage in the continental USA 1900-2017, adjusting reported impact for inflation, per-capita wealth, and the population of affected counties (Collins and Lowe, 2001; Pielke et al., 2008). Due to a lack of global time series of wealth data, reported damage is normalized by means of a GDP scaling. This is based on a less prerequisite approach applied in Munich Re's NatCat, where recorded damages are normalized proportional to regionalized GDP (Munich Re, 2018). This normalization approach assumes that timeseries in current GDP serve as a first order approximation of economic development, implicitly accounting for inflation, changes in wealth per capita and population. To obtain estimates of normalized reported damage per event $E$, reported damage (RD) is scaled proportional to the affected country's change in GDP between the year of occurrence $y$ and the year 2014:

$$NRD_E = RD_E * \frac{GDP_{2014}}{GDP_y}$$ (Equation 2)

We found that GDP scaling removes the significant positive trend from the yearly impacts in the USA (p-values of 0.04 before and 0.14 after normalization). This is in agreement with the findings of existing normalization studies for past TC impacts in the USA (Pielke et al., 2008; Weinkle et al., 2018).

**2.3 Methods**

**2.3.1 Event Matching: Assigning reported damage data to simulated TC events**

For the comparison of simulated and reported TC damage, reported events from EM-DAT per TC and country need to be assigned to TC tracks from IBTrACS. Tracks were matched based on the country affected and timestamps (Lüthi, 2019): (1) In a first step, the impacted countries per TC track is determined, i.e. in which countries a storm does make landfall. (2) Subsequently, the best fitting tracks are assigned to the reported events, based on an iterative comparison of start dates provided in the datasets. Given that countries are hit by several TCs in a relatively short time, the assignment certainty varies. Finally, (3) tracks with a low assignment certainty are double checked manually for removal or re-assigning. In total, we matched 848 EM-DAT events to their respective tracks. These events account for 913 billion USD reported economic damages out of the total 959 billion USD from the 991 EM-DAT events (95%). For 534 of the 848 assigned events, there is an economic damage larger than zero simulated in CLIMADA with the respective TC track. Generally, the difference between simulated and reported damage per matched event span several orders of magnitude. Extreme outliers are likely to be associated either to a mismatch or flawed values of reported damage. Therefore, we exclude 61 extreme outliers from calibration, i.e. all events that come with a deviation of more than factor 1'000 between normalized reported damage and damage simulated with the default impact function.

Eventually, a total of 473 assigned events remain for analysis, referred to as 'matched events' in the following. These matched events, representing damage per TC and country, are based on 376 TC tracks making landfall in 53 countries (One TC can make landfall in several countries). The total reported damage from these 473 matched events accounts to 91% of the sum of all TC-related reported damages from 1980 to 2017 in EM-DAT (76% after normalization). Damage simulated for the 376 TCs with the default impact function amount to 58% of the total global simulated damage from all 4'098 TC tracks.

**2.3.2 Damage Ratios: EDR and TDR**

For the analysis of regional differences in TC vulnerability, event damages are simulated with CLIMADA for all matched events with the default impact function (Sect. 2.4). The event damage ratio (EDR) is computed per matched event $E$ as the ratio of simulated event damage (SED) over normalized reported damage (NRD):

$$EDR_E = \frac{SED_E}{NRD_E} \qquad\qquad \text{(Equation 3)}$$

An EDR of 1.0 indicates a perfect fit between SED and NRD. An EDR greater (smaller) than 1.0 indicates an overestimation (underestimation) of the simulations as compared to reports. As there are considerable deviations between the distribution of EDRs between countries, the median of EDR per country is used to define calibration regions in Sect. 2.3.3.

To compare the aggregated damage on a global or regional level, we use total damage ratio (TDR) defined as the sum of simulated damages divided by the sum of normalized reported damages:

$$TDR_R = \frac{\sum_{E=1}^{N} SED_E}{\sum_{E=1}^{N} NRD_E} \qquad\qquad \text{(Equation 4)}$$

Where $N$ is the number of matched events $E$ in a region $R$.

The distribution of EDR and TDR before calibration as well as TDR after calibration is shown per region in Figures 6 and S4 and per country in the supplementary Figure S2.

**2.3.3 Calibration of regional impact functions**

As a first step towards regional calibration of the TC impact model, distinct calibration regions were defined based on three criteria regarding (1) geography, (2) data availability, and (3) patterns in damage ratios before calibration: (1) We clustered countries by hemispheric ocean basins. This results in five high level regions: North Atlantic and East Pacific oceans (NA),

North Indian Ocean (NI), Oceania (OC), South Indian Ocean (SI), and North West Pacific (WP). This first geographical separation is applied to account for differences in TC characteristics and data sources between the ocean basins (Schreck et al., 2014). The five basins are then subdivided based on (2) a minimum desired number of 30 data points (matched TC events) per region; and (3) the median EDR per country. Applying criteria 2, three countries come with a sufficient amount of data points to be calibrated for themselves: China (N=69), the Philippines (N=83), and the USA (N=43, including three

events in Canada). Applying criterion 3, the remaining countries in WP are further subdivided into two regions: South East Asia with median EDR<1.2 and the rest of the North West Pacific with EDR>5 (see Fig S2d in the Supplement). In summary, the nine calibration regions are the Caribbean with Central America and Mexico (NA1), the USA and Canada (NA2), North Indian Ocean (NI), Oceania with Australia (OC), South Indian Ocean without Australia (SI), South East Asia (WP1), the Philippines (WP2), China mainland (WP3), and the North West Pacific (WP4) (see Fig. 3 and Table A1).

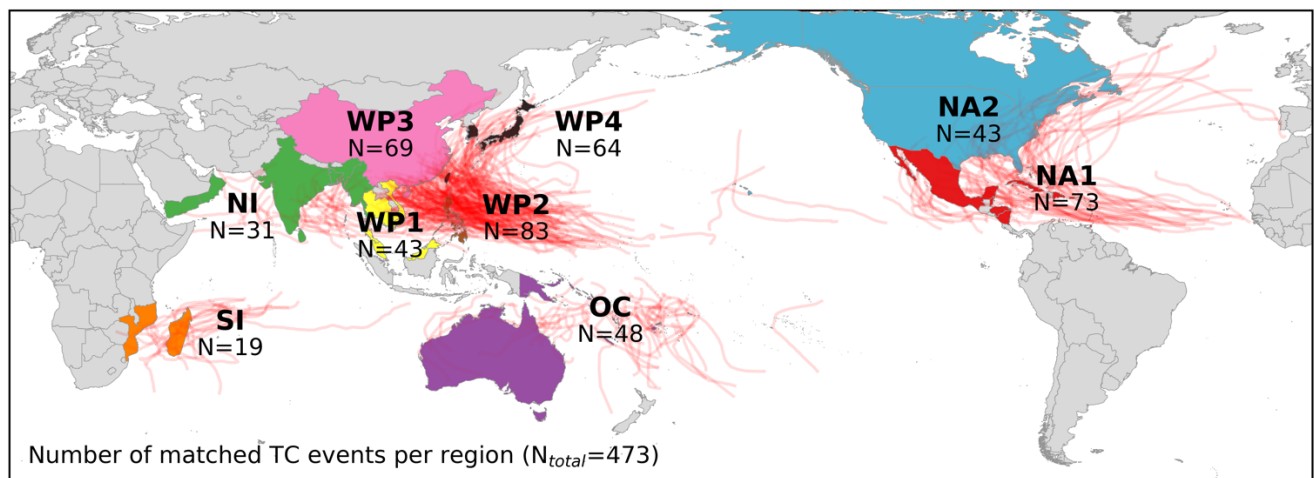


**Figure 3: World map highlighting the 53 countries used for calibration, color coded per calibration region. The tracks of 376 TCs used for calibration are plotted as red lines. The number of resulting matched events N is displayed per region. Regions by color: red: the Caribbean with Central America and Mexico (NA1); blue: the USA and Canada (NA2); green: North Indian Ocean (NI); purple: Oceania with Australia (OC); orange: South Indian Ocean (SI); yellow: South East Asia (WP1), brown: the Philippines**
**(WP2), rose: China Mainland (WP3); black: rest of North West Pacific Ocean (WP4). The countries per region are listed in Table A1.**

Regional impact functions are calibrated following two complementary approaches, based on (1) minimizing the spread of EDR and (2) the optimization of TDR. For the first calibration approach, the root-mean-squared fraction (RMSF) is

introduced as a cost function:

$$RMSF = exp\left(\sqrt{\frac{1}{N}\sum_{E=1}^{N}[\ln{(EDR_E)}]^2}\right)$$ (Equation 5)

Input variables are the number of events $N$ and the natural logarithm of EDR (c.f. Eq. 3). The RMSF is a measure of the spread in EDR, i.e. the relative deviation between modelled and reported damage for all matched events in a region. In the computation of RMSF, each event $E$ has the same weight, independent of the absolute damage values. The natural logarithm

ensures that an overestimation is penalized the same as an underestimation. RMSF is optimized by identifying the impact function associated to the lowest value of RMSF. A value of 1 would indicate perfect fit of all events. For the second calibration approach, TDR is optimized. A TDR larger than 1 implies that the summed simulated damage exceeds the reported values and vice versa. Therefore, TDR is optimized by identifying the impact function associated to a TDR as close to 1 as possible. As TDR is a ratio of damage aggregated over several events, the TDR approach is biased towards better

representing events with large absolute damage values. In both calibration approaches, the slope of the generic impact function (Fig. 2) is calibrated by fitting the parameter $V_{half}$ in Equation 1. An increase in $V_{half}$ corresponds to a flattening of the function and thus lower resulting simulated damage (c.f. Fig. 2). For the fitting of $V_{half}$, damage is simulated for all matched events and an array of $V_{half}$ ranging from 25.8 ms$^{-1}$ to 325.7 ms$^{-1}$ in increments of 0.1 ms$^{-1}$. For each increment,

EDR is computed for all matched events. Consequently, the values of the cost functions RMSF and TDR are computed for
each region and increment of $V_{half}$. Subsequently, the value of $V_{half}$ associated to optimal results for each cost function is
identified. $V_{half}$ optimized per region is used to calculate fitted impact functions per region. The calibrated impact functions
are used to compute the annual average damage (AAD) per region, allowing for the comparison of results with other studies
in Section 3.3.

## 3 Results

### 3.1 Damage ratio with default impact function

The comparison of TC damage simulated globally with a default impact function (Eq. 1 with $V_{half}$ = 74.7 $ms^{-1}$) reveals (1)
inter-regional differences and (2) considerable uncertainties in CLIMADA's ability to reproduce the reported damage values
per event. The distribution of uncalibrated EDR per region is shown in Figure 4. EDR per matched event is shown in Figure
A1, the distribution of EDR per country is shown in Figure S2 in the Supplement.

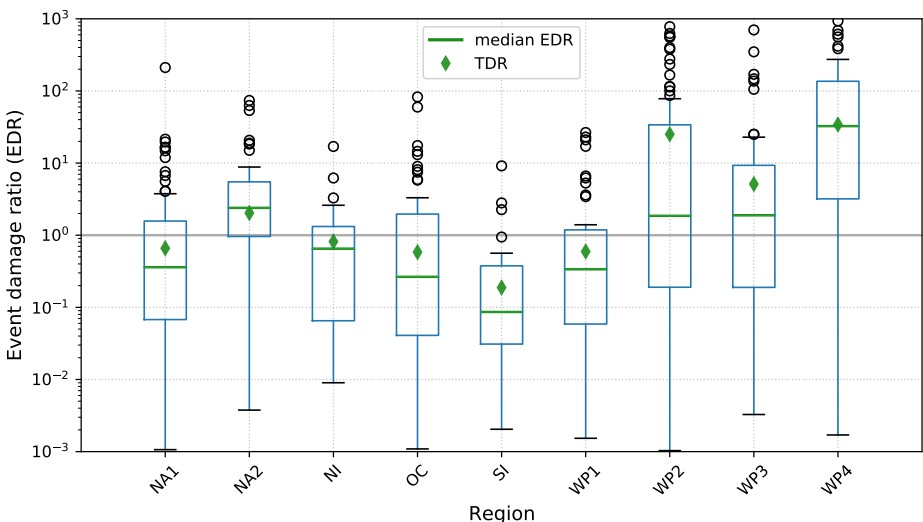


**Figure 4: Spread of event damage ratio (EDR, boxplot) and total damage ratio (TDR) per region before calibration ($V_{half}$=74.7 $ms^{-1}$) per region. The plots are based on data from 473 TC events affecting 53 countries. The EDR boxplots show the median (green line), the first and third quartiles (IQR, blue box), data points outside the IQR but not more than 1.5·IQR distance from either the first or the third quartile (black whiskers), and outliers (black circles). The additional markers show TDR before calibrated (green**
**diamond). The regions are the Caribbean with Central America and Mexico (NA1); the USA and Canada (NA2); North Indian Ocean (NI); Oceania with Australia (OC); South Indian Ocean (SI); South East Asia (WP1), the Philippines (WP2), China Mainland (WP3); rest of North West Pacific Ocean (WP4).**

### 3.1.1 Inter-regional differences

Both the ratios EDR and the cost functions RMSF and TDR show inter-regional differences with regard to the deviation of
the damages simulated with the default impact function from reported damages (Fig. 4 and 6). For most regions, total
simulated and normalized reported damage deviates less than one order of magnitude (Table A2). The outliers are the
regions WP4 (TDR=35.6; Hongkong, Japan, Macao, South Korea, Taiwan) and WP2 (TDR=25.9; the Philippines). For those
two regions, the large value of TDR reveals a mean overestimation of simulated damage as compared to reported damage. In
regions with TDR<1, the uncalibrated model potentially underestimates the damages caused by TCs. These regions are the
Indian Ocean (SI and NI), South East Asia (WP1), Oceania with Australia (OC), and the Caribbean (NA1). The region SI
(Madagascar and Mozambique) shows the overall lowest TDR of 0.2, indicating an underestimation of damages by a factor
of 5.

### 3.1.2 Intra-regional uncertainties

The EDR values within each region show a large spread over several orders of magnitudes (Fig 4). There is no significant correlation between EDR and NRD (Fig. A3), suggesting that the over- and underestimation of simulated event damages is not related to TC severity. The largest spread, as expressed by the RMSF, can again be found in the regions WP4 and WP2 (Fig. 6c). The lowest RMSF was found for the regions NI, NA2, and NA1, i.e. the North Indian and North Atlantic basins. While the large inter-regional differences show the need for a regional calibration of impact functions, the spread of EDR

within some regions point towards uncertainties and limitations of the modelling setup that will not be removed by calibrating the impact function alone.

### 3.2 Regional impact functions

We calibrated regional impact functions to address inter-regional differences in TDR. The resulting impact functions

calibrated with two complementary approaches are shown in Figure 5. The resulting impact functions vary between the regions, both in slope and level of uncertainty, with $V_{half}$ ranging from 46.8 to 190.5 ms$^{-1}$ (Fig. 6a and Table A2). Additional to the regional impact functions, global impact functions were fitted based on all 473 data points combined, resulting in $V_{half}$ ranging from 73.4 (RMSF optimization, i.e. RMSF=min.) to 110.1 ms$^{-1}$ (TDR optimization, i.e. TDR=1). Applying the regional impact functions, TDR calculated for all regions combined is 4.7 for the default impact function and 2.2 for the

RMSF optimized impact functions (Fig. 6b). With the calibration based on TDR optimization, the bias in aggregated simulated damages can be removed, i.e. an impact function is fitted that leads to TDR=1. This does not mean that the simulated damage of each single event is equal to the reported damage. In fact, there is a large spread in the values of $V_{half}$ that would fit best for individual events. This uncertainty is visualized by the interquartile range of the array of impact functions fitted to the individual events per region (shading in Fig. 5). For the individual fitting per event, the value of $V_{half}$ is

determined by what would be required to obtain an EDR equal to 1. The sensitivity of TDR and RMSF per region to changes in $V_{half}$ is visualized in the Supplement: Regions with a large uncertainty, i.e. a large spread of EDR, generally show a relatively low robustness of the cost functions (Fig. S3). On a globally aggregated level, calibration reduces the spread of EDR to a certain degree, placing more than half of events in the EDR range from 10$^{-1}$ to 10.

The comparison of complementary calibration approaches gives an indication of the robustness of the calibration per region. In all regions, the calibrated impact functions based on both approaches lie within the interquartile range of the individually fitted curves (Fig. 5). However, the difference between $V_{half}$ for the two approaches ranges from 3 ms$^{-1}$ (region NA2) to 104 ms$^{-1}$ (WP2). The largest uncertainties were found in the fitting of $V_{half}$ for regions WP2-4 in the North West Pacific. In these regions, the TDR optimization fits values of $V_{half}$ that are much larger than for the RMSF optimization (Fig. 6a). This

corresponds to rather flat impact functions as shown in the bottom row of Figure 5. Since TDR gives larger weight to events with large damage values, these results indicate that these events are systematically overestimated by the model in the regions WP2-4. The flat calibrated impact functions partly compensate this overestimation. As a further indication of large uncertainties, TDR optimization in these three regions returns RMSF values that are larger than with the uncalibrated impact function (Fig. 6c). Possible reasons for the uncertainties in the model are explored in a case study for the Philippines in Sect.

4 and further discussed in Sect. 5.

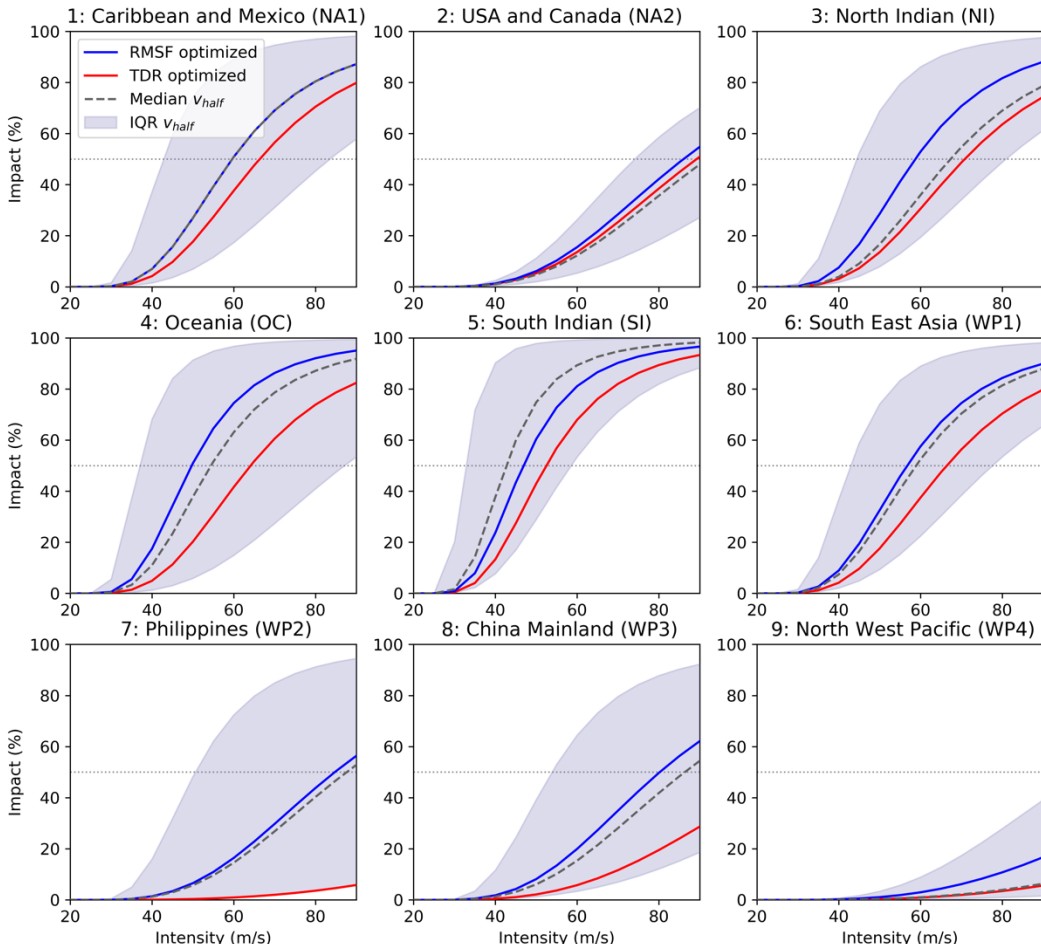

**Figure 5: Regional impact functions for nine calibration regions, based on complementary calibration approaches: RMSF optimized (blue), TDR optimized (red), and the median $V_{half}$ obtained from fitting impact functions for each individual event to obtain an EDR of 1 (dashed). The shading demarcates the range containing 50% of the individually fitted impact functions per region, i.e. the interquartile range (IQR).**


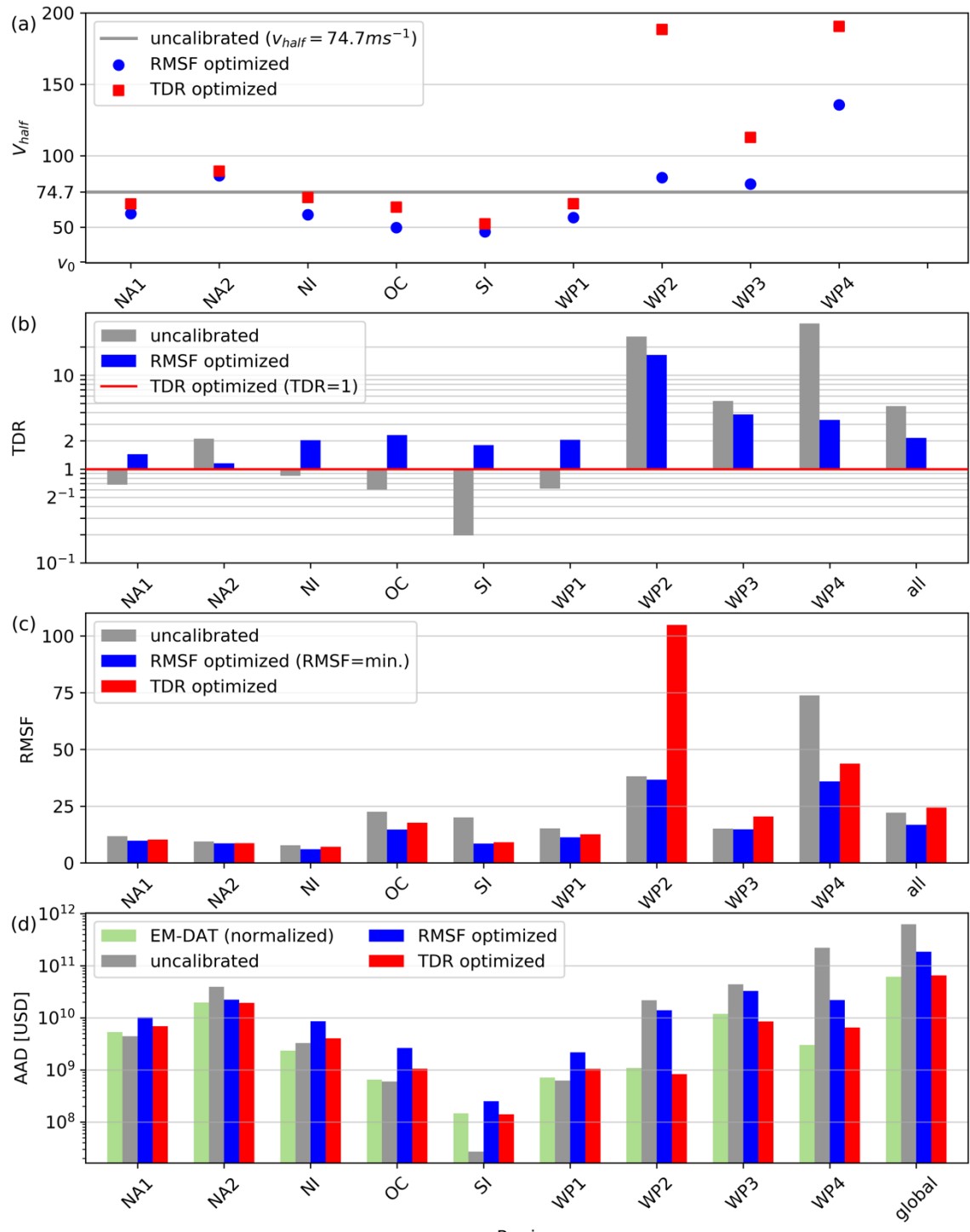

**Figure 6: Calibration results and cost functions for nine calibration regions and all regions combined, each shown before (grey) and after calibration (blue and red): (a) $V_{half}$: fitted impact function parameter; (b) TDR: ratio of total simulated and normalized reported damage; (c) RMSF: root-mean-squared fraction; and (d) AAD: normalized reported (green) and simulated annual expected damage (AAD). AAD is computed from all events available in EM-DAT (N=1650, green) and IBTrACS (N=4098), not just the 473 matched events used for calibration (a-c). Please refer to Tables 1 and A2 for numerical values. The regions are the Caribbean with Central America and Mexico (NA1); the USA and Canada (NA2); North Indian Ocean (NI); Oceania with Australia (OC); South Indian Ocean (SI); South East Asia (WP1), the Philippines (WP2), China Mainland (WP3); rest of North West Pacific Ocean (WP4).**

### 3.3 Annual average damage AAD

Despite considerable interannual variability of TC occurrence and impacts, AAD is often used as a reference value for the mean risk per country or region. Here, we compare AAD computed with the regionalized TC impact model to values from EM-DAT and literature (Table 1). AAD from EM-DAT represents values normalized to 2014, based on all 991 damaging events reported in the database from 1980-2017. Based on the calibrated impact functions, direct damage is simulated based on the full set of TC tracks (N=4'096) and all countries. AAD values per country are provided in the Supplement. The computation of global AAD considers all countries, not only those used for calibration. Thereby, the regionally calibrated impact functions are used for other countries in the same region (c.f. Table A1). AAD in countries not attributed to any region is calculated with impact functions calibrated globally. The resulting AAD for the calibration regions and the global aggregate are shown in Figure 6d and Table 1. The standard deviation of AAD is generally of the same order of magnitude as AAD (Table 1).

For the years 1980 to 2017, we find aggregated global AAD to range from 51 up to 121 billion USD (current value of 2014). In comparison, global AAD from EM-DAT is 46 billion USD. Values from GAR 2013 and Gettelman (2017) range from 67.0 to 88.9 billion USD. It should be noted, however, that the two studies consider different time periods than our study (1950 to 2010 and 1979 to 2012, respectively), as well as deviant TAVs per country. Global TAV for 224 countries aggregates to 251 trillion USD, as compared to 156 trillion USD in Gettelman et al. (2017) and only 96 trillion USD in GAR 2013 (Table 1). Therefore, the comparison of AAD relative to TAV is a better measure to compare the results of the three studies. Relative to TAV, simulated global AAD amounts to 0.2-0.5‰ in our calibrated model, as compared to 0.4-0.5‰ in Gettelman et al. (2017) and 0.9‰ in GAR 2013 (Table 1).

The aggregated region with the largest simulated AAD is East Asia (WP, 17-71 billion USD), followed by the USA with 19-22 billion USD and the North Indian Ocean with 4-9 billion USD. The regions WP2 and WP4 show the largest discrepancy in AAD simulated with the two alternative calibrated impact functions. This is consistent with the large uncertainties found for these regions during calibration (Sect. 3.1 and 3.2). In the most southern regions NI, SI, OC, and WP1, simulated relative AAD is consistently larger than in GAR 2013. This indicates that the calibration corrects for a systematic underestimation of TC vulnerability in these regions. For the Philippines (WP2), the largest AAD relative to TAV was simulated (22.3‰ with the RMSF optimized impact function). While the damage estimates simulated for WP2 come with large uncertainties, the range of relative AAD (1.3-22.3‰) entails the 11.0‰ for the Philippines in GAR 2013. The case of the Philippines will be further analyzed and discussed in the following Sect. 4.

Table 1: Annual average damage (AAD) from calibrated CLIMADA, as well as AAD from EM-DAT (normalized to 2014), GAR 2013 and Gettelman et al. (2017). Total AAD and the standard deviation of annual damage (in brackets) per region is given in current billion USD ($B). AAD relative to total asset value (TAV, c.f. Table A3) is provided in permille (‰). TAV values per region and study are reported in Table A3. Please note that both GAR 2013 and Gettelman et al. (2017) included synthetic TC tracks in their analysis, which are based on historical tracks. The last row (world) considers all countries. AAD values by country are provided in the Supplement.

*) USA and Bermuda, without Canada.

| Region | AAD EM-DAT | AAD Calibrated CLIMADA: RMSF optimized | | AAD Calibrated CLIMADA: TDR optimized | | AAD GAR 2013 | | AAD Gettelman et al. (2017) | |
|---|---|---|---|---|---|---|---|---|---|
| | $B (2014) | $B (2014) | ‰ of TAV | $B (2014) | ‰ of TAV | $B (2005) | ‰ of TAV | $B (2015) | ‰ of TAV |
| NA1 | 5.3 (14.2) | 10.3 (16.1) | 2.2 | 6.9 (11.7) | 1.5 | 4.6 | 2.1 | 9.5 (17.8) | 0.3-1.1 |
| NA2 | 19.7 (43.1) | 22.4 (32.5) | 0.4 | 19.4 (28.2) | 0.3 | 11.8 | 0.5 | 11.0 (15.5) | 0.2* |
| NI | 2.3 (3.8) | 8.6 (13.9) | 1.4 | 4.1 (6.7) | 0.6 | 0.3 | 0.2 | | |
| OC | 0.7 (0.8) | 2.6 (3.6) | 0.4 | 1.1 (1.6) | 0.2 | 0.1 | 0.1 | | |
| SI | 0.1 (0.3) | 0.3 (0.6) | 5.7 | 0.1 (0.4) | 3.2 | 0.0 | 2.8 | | |
| WP1 | 0.7 (1.2) | 2.2 (3.4) | 1.0 | 1.1 (1.6) | 0.5 | 0.0 | 0.0 | | |
| WP2 | 1.1 (1.8) | 14.0 (34.6) | 22.3 | 0.8 (2.3) | 1.3 | 2.0 | 11.0 | | |
| WP3 | 11.9 (14.8) | 32.9 (39.4) | 1.0 | 8.6 (10.3) | 0.3 | 9.0 | 2.0 | | |
| WP4 | 3.0 (4.0) | 21.9 (24.3) | 0.8 | 6.6 (7.3) | 0.2 | 60.0 | 3.1 | | |
| Σ WP | 16.8 | 71.0 | 1.2 | 17.0 | 0.3 | 71.1 | 2.8 | 61.4 (53.8) | 0.9-1.0 |
| Σ all | 45.0 (54.8) | 115.2 (72.4) | 0.8 | 48.6 (33.2) | 0.3 | 87.9 | 1.6 | | |
| **World** | **46.3 (55.6)** | **120.9 (73.9)** | *0.5* | **50.6 (33.6)** | *0.2* | **88.9** | *0.9* | **84.6 (63.9)** | *0.4-0.5* |

## 4. Explorative case study: the Philippines

For a better understanding of the uncertainties involved in the TC impact function calibration, we exploratively examine simulated and reported damages of matched events in the Philippines (region WP2). The Philippines is the region with the least robust calibration results, with a large spread in EDR and the largest discrepancy between the two calibration approaches: The difference in $V_{half}$ between the two calibration approaches exceeds 100 ms$^{-1}$ (Fig. 6a). Consequently, there is a large spread in simulated AAD, ranging from 0.8 to 14 billion USD (Table 1). This corresponds to an underestimation of annual risk by 0.3 billion USD up to an overestimation by 21.2 billion USD as compared to normalized values from EM-DAT with an AAD of 1.1 billion USD.

The goal of this explorative case study is to better understand what drives these uncertainties in the TC impact model within the region, discuss the limitations of the calibrated model, and identify points for improvement for the future development of global TC impact models. Thereby, we assess the following hypotheses: (1) Potential differences between urban and rural exposures and vulnerabilities as considered in the GAR 2013 (De Bono and Mora, 2014) are not fully resolved in the model. (2) The simplified representation of the TC hazard intensity with wind speed alone is not capable to adequately model the impact of TCs with over-proportional damage caused by sub-perils like storm surge and torrential rainfall (Baradaranshoraka et al., 2017; Park et al., 2013). In the following, we explore these hypotheses by the example of 83 matched TC events in the Philippines, while keeping in mind that the model setup is not designed to represent single events perfectly, due to the large inherent stochastic uncertainty. To explore these hypotheses, we review reports and literature on TC impacts in the Philippines, and examine the relationship between EDR per event with the spatial distribution of the wind field and subsequent simulated damages associated to each single event.

## 4.1 Tropical cyclones in the Philippines

The Republic of the Philippines is one of the most TC-prone countries in the world (Blanc and Strobl, 2016). From 1951 to 2014, an annual average of 19.4 TCs entered the Philippine Area of Responsibility (Cinco et al., 2016), with six to nine TCs making landfall in the Philippines each year (Blanc and Strobl, 2016; Cinco et al., 2016). This is a relative high frequency compared to five to eight landfalls in China (Zhang et al., 2009), and an average of three landfalls per year in the North Indian Ocean region (Wahiduzzaman et al., 2017) as well as in the USA (Lyons, 2004). The north and east of the Philippines are the regions most exposed to TC landfalls, with most TCs crossing the Philippines from east to west (Cinco et al., 2016; Espada, 2018). Rainfalls associated to TCs contribute around 35% of annual precipitation in the Philippines, with regional values ranging from 4 % to 50 % (Cinco et al., 2016).

In total, 83 matched TCs making landfall in the Philippines were used for calibration. For 11 of the 21 most damaging TC events, reports and scientific literature on associated sub-perils and impacts were reviewed (Table A4). In summary, TCs making landfall in the Philippines cause damage due to large wind speed, storm surge, as well as rain induced floods and landslides. Meteorologically, the storm systems interact with the monsoon season, affecting both dynamics and the severity of torrential rain (Bagtasa, 2017; Cayanan et al., 2011; Yumul et al., 2012). TCs in the Philippines inflict damage on several sectors, most costly on housing and agriculture, but also on schools and hospitals, power and water supply, roads, and bridges (Table A4). Single events were also reported to damage and disrupt airports and ports (Typhoon Haiyan) and dikes (Nesat and Xangsane). This complexity of how and where TCs cause damage in the Philippines is in stark contrast to the relatively simple representation of hazard and exposure in our modelling setup. It is therefore not surprising, that our calibrated TC impact model is over- and underestimating the damage of individual events, as illustrated for the Philippines by the wide spread of EDR. In the following, we will take a closer look at events with over- and underestimated simulated damage to explore the two hypotheses above.

## 4.2 Urban vs. rural exposure

Most of the asset exposure value of the Philippines is concentrated around the metropolitan area of Manila (Metro Manila). Located around 14.5°N, 121.0°E (Fig. 7a), Metro Manila is Philippine's political and socio-economic center (Porio, 2011). The Typhoons Angela (1995), Xangsane (2006), and Rammasun (2014) are prominent TCs hitting the Metro Manila directly. In our analysis, these TCs come with particularly large EDRs, i.e. an overestimation of simulated vs reported damages, even with calibrated impact functions (Table A4). All three typhoons show maximum sustained wind speeds in Manila larger than 50 ms$^{-1}$ (Fig. 7b,e,f), corresponding to relative damage ranging from 6 up to 37 % of asset exposure value with the calibrated impact function. These large relative damages in combination with the concentration of asset exposure value in the Manila region are likely to explain the large EDRs of these events. The analysis of all 83 TC events used for calibration support this hypothesis, underpinning the crucial role the large asset exposure values in the Metro Manila plays for the wind-based damage simulation: An overestimation of simulated damages (e.g. EDR>10) consistently coincides with large wind speeds over Metro Manila : Out of 19 TCs affecting Manila directly, we find 16 (84%) with an EDR>10 and zero occurrences of EDR<0.1 (Fig. 8). In contrast, only 9 of 64 TCs not affecting Manila directly come with an EDR>10. In summary, we found simulated damage of an event more usual to substantially exceed normalized reported damage if the event hit Manila directly. This confirms hypothesis (1) that a special treatment of the impact functions for urban areas could improve the TC impact model.

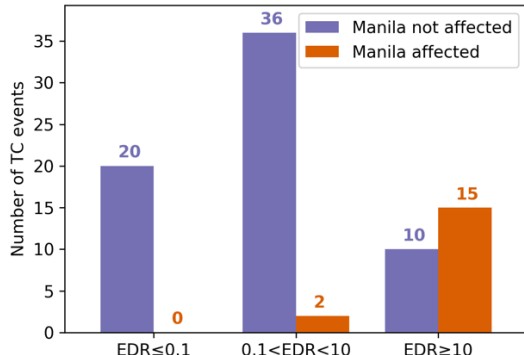

**Figure 7: Maps of the Philippines showing (a) the spatial distribution of asset exposure value in the Philippines [current US dollar of 2014] based on Eberenz et al (2020); and (b-f) mapped TC impacts for Typhoon Rammasun (b), Typhoon Haiyan (c), Tropical Storm Ketsana (d), Typhoon Xangsane (e), and Typhoon Angela (f). For each event, the map shows the TC track from IBTrACS (bold solid line), the spatial distribution of simulated maximum sustained wind speed in ms$^{-1}$ (dashed lines at 25, 50, and 70 ms$^{-1}$), and simulated direct damage at a 10km resolution (color shading). Coast lines and the location of major cities based on Cartopy (Met Office, 2010).**

**Figure 8: Distribution of the event damage ratio (EDR) for 83 TCs making landfall in the Philippines from 1980 to 2017. The number of events for three ranges of EDR are compared, differentiating whether Manila was directly affected by the TC's wind field (orange) or not (purple). Manila is considered to be affected if the hazard intensity exceeds 25 ms$^{-1}$ at 14.5°N, 121.0°E.**

**4.3 Impact of storm surge and torrential rain**

While urban vulnerability to strong winds in Metro Manila appears to be overestimated by the calibrated impact function, Metro Manila is known to be highly exposed and vulnerable to regular, large scale flooding (Porio, 2011). The main drivers of flood vulnerability are its geographical setup, largely unregulated urban growth and sprawl, and substandard sewerage systems, especially in low-income areas (Porio, 2011). Tropical Storm Ketsana, locally known as Ondoy (2009) is an example with very low simulated damages coinciding with large reported damages associated to flood in Metro Manila: Ketsana's EDR is 0.002, i.e. simulated damage is more than two orders of magnitude smaller than reported. The large reported damage (NRD=401 million USD) was mainly due to floods and landslides. Torrential rainfall caused severe river flooding in Metro Manila and landslides around Baguio City, resulting in severe damages (Abon et al., 2011; Cruz and Narisma, 2016; Nakasu et al., 2011; NDCC, 2009a). The flood damages were not resolved by the wind-based impact model, with intensities well below 50 ms$^{-1}$ and neither affecting Manila nor the northern Baguio City directly (Fig. 7d). Notably, even for TCs with large overestimation of simulated damage due to high wind speeds in Metro Manila, namely Fengshen and Xangsane, a substantial part of the reported damage was actually caused by pluvial flooding and landslides and not by wind alone (Yumul et al., 2008, 2011, 2012).

For the most severe TC in the recent history of the Philippines, Typhoon Haiyan (2013), normalized reported damage and simulated damage are in the same order of magnitude resulting into an EDR of 0.17. Haiyan, with sustained 1-min surface wind speeds up to 87.5 m/s, caused thousands of casualties and around 10 billion USD of economic damage in the Philippines (Guha-Sapir, 2018; Mas et al., 2015). Devastating wind and storm surge associated to Haiyan caused damage to multiple sectors, including ports and an airport. It should be noted that sector specific impacts are not resolved in the impact model and Haiyan did not affect Manila directly. Relatively large damages were simulated around Tacloban City, Leyte, which was actually devastated by Haiyan's storm surge. Large wind impacts were also simulated further West around the cities Iloilo and Cebu (Fig. 7c) that were not as exposed to surge as Leyte province. The relatively good performance of the model in the case of Haiyan is thus not explained by a perfect location and representation of the impact in the model. It is rather based on overestimated urban wind damages partly balancing the lack of damages caused by storm surge.

**4.4 Conclusions from the case study**

The case of the Philippines reveals limitations of the model and calibration due to the lack of an explicit representation of sub-perils such as storm surge, torrential rainfall, and landslides (Sect. 4.3). The flood damage caused by Ketsana is a showcase example for severe damages associated with a TC with relatively low wind speeds, that is, an event that cannot be adequately reproduced with a wind-based impact function. Adding to the stochastic uncertainty, the magnitude of rainfall during a TC events in the Philippines is not only determined by the intensity of the TC event, but also by the coinciding monsoon season, as in the case of the Typhoons Fengshen and Haiyan (Espada, 2018; IFRC, 2009; Yumul et al., 2012).

Next to a lack of representation of all components of hazard intensity, differences in exposure and vulnerability between urban and rural areas exposed to TCs are likely to contribute to the large spread in EDR and subsequently uncertainty in the impact function calibration. This has been illustrated in Sect. 4.2: The large overestimation of simulated event damage of TCs affecting the Manila metropolitan area points towards relevant sources of epistemic uncertainty: On the one hand, a large share of exposed asset values in the model is concentrated in urban areas, while exposed agricultural assets in rural areas are neglected. On the other hand, one single impact function might not be sufficient to represent both urban and rural building vulnerability. Another factor contributing to the high simulated damages in Manila could be the wind field model: Manila is located in a bay on the west coast of the main island Luzon. Most TCs are approaching Luzon from the east. The wind field model adapted from Holland (2008) does however not take into account variation in topography and surface roughness. This could lead to an

overestimation of simulated wind speeds downstream of elevated land, as in the case of Manila. A better representation of wind speed over land could mitigate this problem (Done et al., 2019).

## 5 Discussion

### 5.1 Relevance for TC risk assessments

In this study, we showed how the regionalization of impact functions improves the assessment of TC risk in numerous world regions, correcting an overestimation of aggregated TC damages by a factor of potentially up to 36 in the North West Pacific, and an underestimation by the factor 5 in the South Indian Ocean. To complement the global perspective, we explored the limitations of the TC impact modelling setup by the case study of TC events in the Philippines.

The calibration resulted in large regional differences in the slope of impact functions, with considerable consequences on the magnitude of simulated damages. In Sect. 3.2, we compared average damages simulated with regionalized impact functions to results from literature. While the comparison is limited by differences in the model setups, we found that regional damage estimations relative to the exposed asset values generally agree well to the results of previous studies. However, the results for the North West Pacific region (WP4), consisting of Japan, South Korea, Macao, Hongkong, and Taiwan, deviate substantially from GAR 2013. Simulated relative AAD in the region ranges from 0.2-0.8 ‰ as compared to 3.1 ‰ in GAR 2013. This difference implies that, besides the use of building type specific impact functions, the TC impact model of GAR 2013 substantially overestimates TC damages in WP4 compared to reported data. Consistent with this finding, the uncalibrated simulation showed the largest overestimation of aggregated damages in this region. Assuming that the order of magnitude of reported direct damages from EM-DAT is reasonable, the regionalization of impact functions presented here is an improvement for TC risk assessments in the region.

For calibration, two complementary approaches were employed: The optimization of aggregated simulated compared to reported damages (TDR), and the minimization of the spread of damage ratios of single events (RMSF).
Annual average damage simulated based on the TDR optimized set of impact functions are generally closer to the values found in EM-DAT than the values based on RMSF optimization. This is not surprising, since TDR is designed to represent aggregated damage per region. For the assessment of TC risk on an aggregated level, it is therefore most appropriate to employ the more conservative TDR optimized model, even though single events can be massively underestimated with the flatter impact functions. Complementary, impact functions based on RMSF optimization and the spread of individually event fitting can be included in risk assessments for sensitivity analysis.

### 5.2 Uncertainties and limitations

The deviation between the results of the two calibration approaches indicates how robust the calibration is with regards to the model's ability to represent the correct order of magnitude of single event damage. Whereas the model setup returns reasonable risk estimates and consistent calibration results for Central and North America, we found an extensive spread in EDR and calibration results for other regions, especially in East Asia. While the correlation between simulated and reported event damages is improved by the calibration, the simulated damage of single TC events can deviate several orders of magnitude from reported damages (Fig. 4, A1 and A2). In the regions of the North West Pacific (WP2-4), the fitted impact functions are ambiguous, with large discrepancies between the two calibration approaches. The low robustness found for these regions stems from multiple causes, including the stochastic uncertainty in TCs as natural phenomena, as well as

epistemic uncertainties located in the hazard, exposure, and vulnerability components of the impact model. An additional source of uncertainties is located in the reported damages used for reference. Future improvement of the TC impact model and a sound judgement of the limitations of the calibrated impact functions requires better understanding of the epistemic

uncertainties. In the following, we will discuss these uncertainties for the different components of the model.

The case of the Philippines provides insights into the uncertainties located in the model setup, both in the representation of hazard intensity and in differences between the structure and vulnerability of exposed assets in urban and rural areas (Sect. 4). The hazard is represented by wind fields modelled from TC track data and the same impact functions are applied in urban

and rural areas. These are considerable simplifications of the actual interaction of cyclones with the natural and built environment. To reduce these uncertainties, the hazard component could be improved by considering topography (Done et al., 2019) and complementing wind speed with sub-perils like storm surge, torrential rain, and landslides. For a better representation of urban assets, building type specific impact functions, and a differentiation of urban and rural exposure as applied for GAR 2013 (De Bono and Mora, 2014) could be beneficial. Furthermore, geospatial agricultural yield data could

be added to the exposure data, albeit reported damage for calibration is mostly not available at such sectoral granularity. Next to the model setup, the reported damage data obtained from EM-DAT are another relevant source of uncertainty. Reported damage data are expected to come with considerable uncertainties, partly due the heterogeneity of data sources, the blending of direct and indirect economic damages, as well as  political and structural reporting biases (Guha-Sapir and Below, 2002; Guha-Sapir and Checchi, 2018). Further uncertainty is introduces by the lack of international standards for

reported damage datasets, leading to inconsistencies between data providers (Bakkensen et al., 2018b). These uncertainties limit our understanding of the robustness of the calibration. For future calibration studies relying on reported damage data, calibration robustness could be increased by combining datasets from different sources in an ensemble of datasets (see Zumwald et al., 2020).

In this study we did not explicitly quantify the uncertainties related to the model setup, the input data for hazard and exposure, as well as the reported data used as reference data for calibration. Rather, the robustness of the calibrated impact functions was judged based on the deviation between the two calibration approaches and the spread of impact functions fitted to the individual TC events. Based on the limitations discussed above, we conclude that the resulting array of regionalized impact functions should be applied with caution, being aware that the model setup is not suitable to represent

single TC events adequately. However, the calibrated impact functions mark an improvement for the modelling of aggregated risk estimates, such as the annual average damage. Impact functions sampled from the range of calibration results can be applied for a more probabilistic modelling of TC impacts. It should also be noted that the impact functions calibrated for the years 1980-2017 cannot be expected to be stable in the future. Applying these impact functions for the assessment of future TC risk requires a ceteris paribus assumption with regard to vulnerability.


While the results of this study are not specific to the CLIMADA modeling framework, the precise shape and scaling of the calibrated impact functions are, however, to a certain degree specific to the choices and input data of the modeling setup: (1) The choice of free parameters in the impact function (c.f. Section 2.2.3 and Lüthi, 2019); (2) The TAVs (c.f. Table A3): impact functions would scale differently with a different assumed inventory of exposed assets; (3) spatial resolution; and (4)

the representation of hazard intensity: The regionalized impact functions presented here were calibrated for wind-based damage modelling on a spatially aggregated level. Model setups with an explicit representation of related sub-perils like storm surge or torrential rain require different (i.e. flatter) impact functions for the wind-induced share of TC damage, as well as additional impact functions for all sub-perils. Likewise, impact models with an explicit representation of building types and agricultural assets require a more differentiated set of impact functions. Considering the irreducible stochastic

uncertainties in the system, it remains to be shown to which degree the large inter-regional differences in calibrated impact functions found in this study can be explained by regional differences in building types and standards, physical TC characteristics, or other factors.

## 6 Conclusion and outlook

In this article, the global assessment of TC risk was improved by regionalizing the vulnerability component of the TC impact

assessment. To better account for regional differences, a TC impact model was calibrated by fitting regional impact functions. The impact functions were calibrated within the CLIMADA risk modelling framework, using reported direct economic damage estimates from the EM-DAT dataset as reference data. For calibration, two complementary optimization approaches were applied, one aiming at minimizing the deviation of single event damages from the reported data and one aiming at minimizing the deviation for total damage aggregated over 38 years of data. By fitting impact functions, we were

able to reduce regional biases as compared to reported damage data, especially for countries in the North West Pacific and South Indian Ocean regions. The substantial over-estimation of TC damages in the North West Pacific with the default impact function opens the question for the drivers of the apparently lower vulnerability in this region. Considering the inability of the model setup to directly represent the impacts from TC surge and pluvial flooding, one would rather expect aggregated calibrated impact functions to be steeper than the default wind impact function. Therefore, we suggest

investigating interregional differences in possible other drivers, including building standards but also damage reporting practices. A study combining the empirical evidence provided by reported damage data on the one hand with socio-economic indicators on the other hand would be desirable but rather challenging, as this would add even more layers of complexity and cascading uncertainties to the calibration, especially on a global level.

The calibrated model comes with considerable uncertainties related both to the impact model setup and the reported damage

data. The largest uncertainties were found for the North West Pacific regions, while the calibration produced consistent results for the North Atlantic regions. The spread of fitted impact functions within each region can be exploited to better account for these uncertainties in probabilistic risk assessments. Based on our findings, we recommend to always consider inter-regional differences in vulnerability for the application in global TC impact models. For model setups comparable to the one described here, we recommend the use of TDR optimized functions for risk assessments on an aggregated level. The

resulting simulated damage can complement reported damage data. Assuming that reported damages are more likely to underestimate actual impacts, it could be advisable to sample impact functions from the range between the complementary calibration results. For probabilistic impact modelling, a random sampling from the array of impact functions fitted to individual events could be considered. This becomes especially relevant for regions with large uncertainties attached to the calibration results, such as the North West Pacific and Oceania. Limitations of our research motivate future work. For TC

impact models, we echo the call for a more refined representation of TC hazard as a combination of wind, surge, and rain induced flood and landslide events. When modeling multiple TC sub-perils, aggregated reported damage data are not sufficient to constrain impact function calibration. This might be resolved by consulting socio-economic and engineering type data and knowledge. Furthermore, our case study for the Philippines suggests that differentiating between urban and rural asset exposure, considering topography in wind speed estimations, and the inclusion of exposed agricultural assets

could further increase model accuracy.

# Appendix A

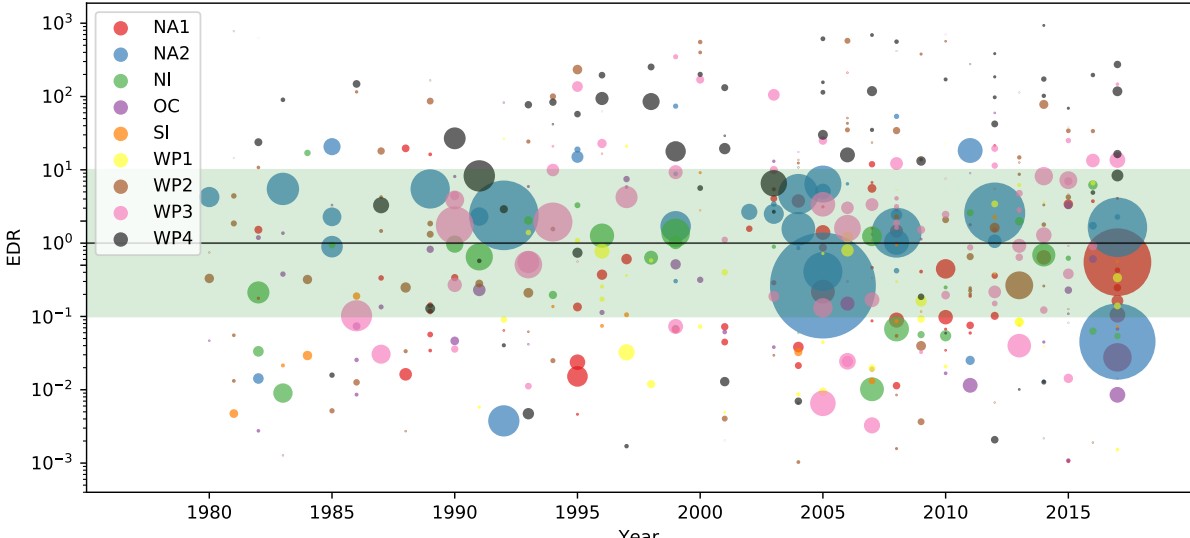

**Figure A1: Event damage ratio (EDR) from 1980 to 2017 for matched 473 TC events worldwide. The nine calibration regions are differentiated by color. The area size of the dots represents the absolute normalized reported damage (NRD) per event. The green shading demarcates the range from EDR=0.1 to 10.**

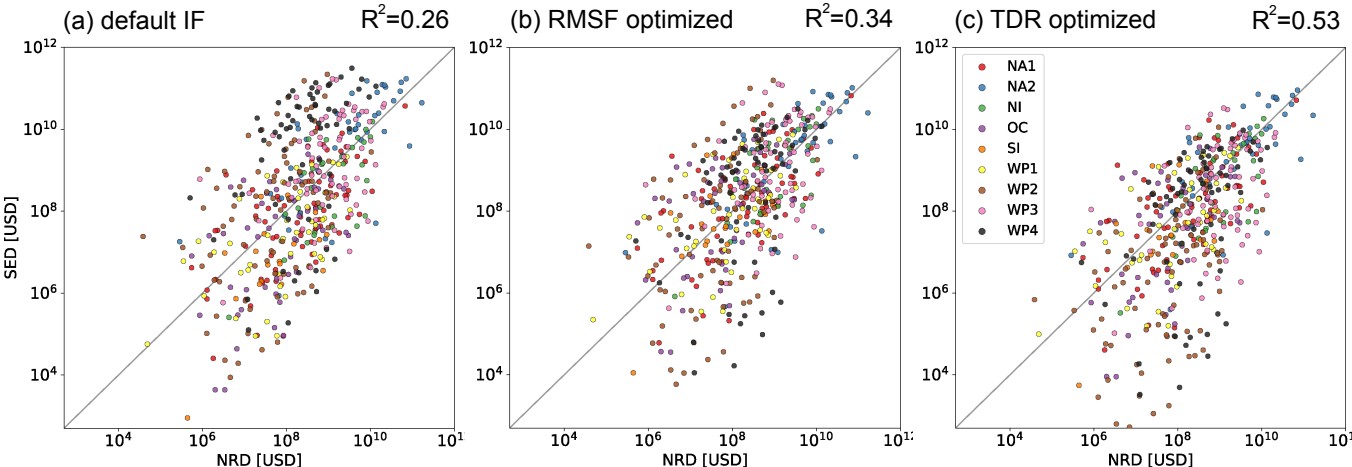

**Figure A2: Simulated event damage (SED) vs normalized reported damage (NRD) for 473 TC events worldwide computed with three different sets of impact functions: (a) uncalibrated default ($V_{half}$=74.7 ms$^{-1}$), (b) RMSF optimized, and (c) TDR optimized. The nine calibration regions are differentiated by color.**

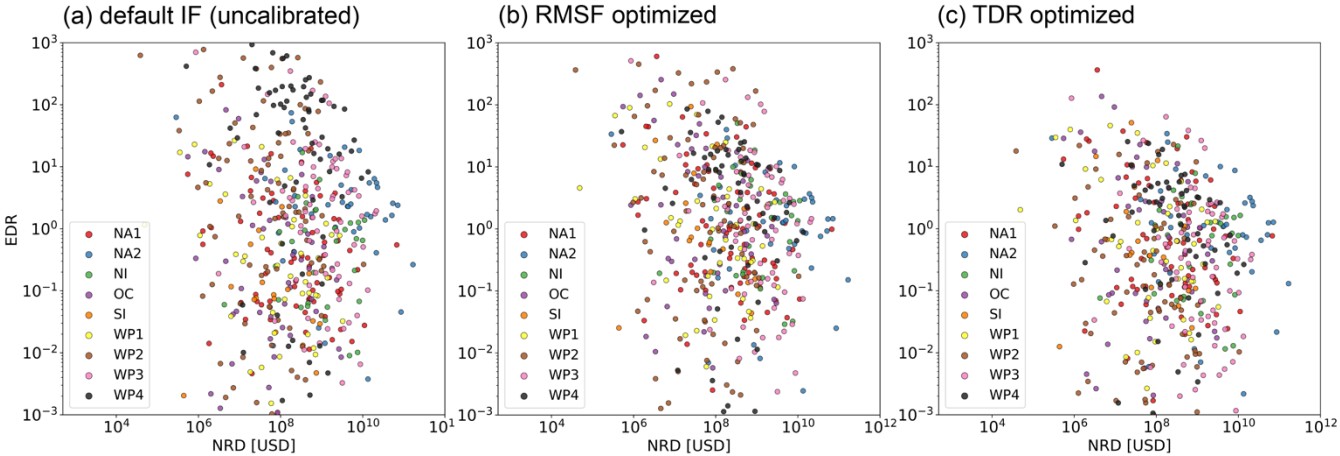

**Figure A3: No significant correlation between event damage ratio (EDR) and normalized reported damage (NRD) was found. The scatter plots show the relationship for 473 TC events worldwide computed with three different sets of impact functions: (a) uncalibrated default ($V_{half}$=74.7 ms-1), (b) RMSF optimized, and (c) TDR optimized. The nine calibration regions are differentiated by color.**

**Table A1: List of countries per calibration region. Countries marked with an asterisk (*) are considered for calibration (53 in total).**

| Region | N countries (calib- ration) | Countries |
|---|---|---|
| North Atlantic 1 (NA1) | 48 (21) | Anguilla; Antigua and Barbuda*; Argentina; Aruba; Bahamas*; Barbados; Belize*; Bermuda*; Bolivia, Plurinational State of; Cabo Verde*; Cayman Islands; Chile; Colombia; Costa Rica; Cuba*; Dominica*; Dominican Republic*; Ecuador; El Salvador; Falkland Islands (Malvinas); French Guiana; Grenada; Guadeloupe; Guatemala; Guyana; Haiti; Honduras*; Jamaica*; Martinique; Mexico*; Montserrat*; Nicaragua*; Panama; Paraguay; Peru; Puerto Rico*; Saint Helena, Ascension and Tristan da Cunha; Saint Kitts and Nevis*; Saint Lucia*; Saint Vincent and the Grenadines*; Sint Maarten (Dutch part); Suriname; Trinidad and Tobago*; Turks and Caicos Islands*; Uruguay; Venezuela, Bolivarian Republic of; Virgin Islands, British*; Virgin Islands, U.S.* |
| North Atlantic 2 (NA2) | 2 (2) | Canada*; United States of America* |
| North Indian (NI) | 36 (6) | Afghanistan; Armenia; Azerbaijan; Bahrain; Bangladesh*; Bhutan; Djibouti; Eritrea; Ethiopia; Georgia; India*; Iran, Islamic Republic of; Iraq; Israel; Jordan; Kazakhstan; Kuwait; Kyrgyzstan; Lebanon; Maldives; Mongolia; Myanmar*; Nepal; Oman*; Pakistan; Qatar; Saudi Arabia; Somalia; Sri Lanka*; Syrian Arab Republic; Tajikistan; Turkmenistan; Uganda; United Arab Emirates; Uzbekistan; Yemen* |
| Oceania (OC) | 26 (11) | American Samoa; Australia*; Cook Islands; Fiji*; French Polynesia*; Guam*; Kiribati; Marshall Islands; Micronesia, Federated States of*; Nauru; New Caledonia*; New Zealand; Niue; Norfolk Island; Northern Mariana Islands; Palau; Papua New Guinea*; Pitcairn; Samoa*; Solomon Islands*; Timor-Leste; Tokelau; Tonga*; Tuvalu; Vanuatu*; Wallis and Futuna |
| South Indian (SI) | 11 (2) | Comoros; Congo, Democratic Republic of the; Eswatini; Madagascar*; Malawi; Mali; Mauritius; Mozambique*; South Africa; Tanzania, United Republic of; Zimbabwe |
| North West Pacific 1 (WP1) | 6 (4) | Cambodia*; Indonesia; Lao People's Democratic Republic; Malaysia*; Thailand*; Viet Nam* |
| North West Pacific 2 (WP2) | 1 (1) | Philippines* |
| North West Pacific 3 (WP3) | 1 (1) | China, Mainland* |
| North West Pacific 4 (WP4) | 5 (5) | Hong Kong*; Japan*; Korea, Republic of*; Macao*; Taiwan, Province of China* |

Table A2: Resulting impact function slope parameter $v_{half}$ and optimization metrics RMSF and TDR per region for (a) the global default impact function (uncalibrated), (b) calibrated by optimizing RMSF, and (c) calibrated by optimizing TDR. The regions NA1 to WP4 are defined in Table A1. The row "combined" summarizes results for all regions combined based on the regionalized calibration; the row "global" is based on one unified global calibration based on all matched TC 473 events. RMSF: root-mean-squared fraction; TDR: total damage ratio.

| Region | Number of | | $V_{half}$ [ms$^{-1}$] | | | RMSF | | | TDR | | |
|---|---|---|---|---|---|---|---|---|---|---|---|
| | countries | events | (a) | (b) | (c) | (a) | (b) | (c) | (a) | (b) | (c) |
| NA1 | 21 | 73 | 74.7 | 59.6 | 66.3 | 11.8 | 9.8 | 10.3 | 0.68 | 1.44 | 1.0 |
| NA2 | 2 | 43 | 74.7 | 86 | 89.2 | 9.5 | 8.7 | 8.7 | 2.11 | 1.16 | 1.0 |
| NI | 6 | 31 | 74.7 | 58.7 | 70.8 | 7.8 | 6 | 7.2 | 0.85 | 2.03 | 1.0 |
| OC | 11 | 48 | 74.7 | 49.7 | 64.1 | 22.5 | 14.7 | 17.7 | 0.6 | 2.31 | 1.0 |
| SI | 2 | 19 | 74.7 | 46.8 | 52.4 | 20.1 | 8.6 | 9.1 | 0.2 | 1.8 | 1.0 |
| WP1 | 4 | 43 | 74.7 | 56.7 | 66.4 | 15.2 | 11.3 | 12.6 | 0.62 | 2.05 | 1.0 |
| WP2 | 1 | 83 | 74.7 | 84.7 | 188.4 | 38.2 | 36.7 | 104.9 | 25.89 | 16.44 | 1.0 |
| WP3 | 1 | 69 | 74.7 | 80.2 | 112.8 | 15.2 | 14.8 | 20.5 | 5.32 | 3.83 | 1.0 |
| WP4 | 5 | 64 | 74.7 | 135.6 | 190.5 | 73.8 | 35.9 | 43.8 | 35.56 | 3.35 | 1.0 |
| combined | 53 | 473 | 74.7 | - | - | 22.2 | 16.8 | 24.4 | 4.69 | 2.15 | 1.0 |
| global calibration | 53 | 473 | 74.7 | 73.4 | 110.1 | 22.2 | 22.2 | 33.1 | 4.69 | 4.84 | 1.0 |

Table A3: Total asset exposure values (TAV) per region. First column: TAV based on Eberenz et al. (2020) as used in this study. Second and third column: Reference values of TAV from GAR 2013 and Gettelman et al. (2017). The unit is $10^{12}$ US dollars ($T) in the current value of the year noted in brackets. AAD relative to TAV is reported in Table 1. *) USA and Bermuda

| Region | TAV Eberenz et al. (2020) | TAV GAR 2013 | TAV Gettelman et al. (2017) |
|---|---|---|---|
| | *$T of 2014* | *$T of 2005* | *$T of 2015* |
| NA1 | 4.66 | 2.19 | 8.6 |
| NA2 | 62.19 | 24.06 | 73.3* |
| NI | 6.32 | 1.64 | |
| OC | 5.94 | 1.85 | |
| SI | 0.04 | 0.01 | |
| WP1 | 2.27 | 0.83 | |
| WP2 | 0.63 | 0.19 | |
| WP3 | 31.40 | 4.51 | |
| WP4 | 26.98 | 19.51 | |
| WP | 61.28 | 25.04 | 58.8 |
| GLB | 250.88 | 96.45 | 155.9 |

**Table A4: Detailed information on the 21 TCs in the matched events list with the largest normalized reported damage: storm name (local name in brackets) and year, normalized reported damage (NRD), simulated event damage (SED, simulated with $V_{half}$=84.7 ms$^{-1}$), event damage ratio (EDR), simulated intensity for the capital city Manila (at 14.5°N, 121.0°E), associated disasters according to literature and EM-DAT as well as affected sectors and asset types as reported by literature. Sources of information: 1) peer-reviewed study; 2) public report 3) data field 'associated disasters' in EM-DAT.**

| Event | SED [mio USD] | NRD[3] [mio USD] | EDR $\left(\frac{SED}{NRD}\right)$ | $V_{wind}$ Manila [m/s] | Associated disasters | Affected sectors & assets | Reference |
|---|---|---|---|---|---|---|---|
| Rammasun (Glenda), 2014 | 39,528 | 821 | 48.17 | 52.8 | Wind[1], Flood[3] | Agriculture[2], buildings (664k)[2], power supply[2], and roads and bridges[2] | (Espada, 2018; NDRRMC, 2014) |
| Haiyan (Yolanda), 2013 | 1,804 | 10,469 | 0.17 | - | Wind[1], Surge[1,2,3] | Agriculture[2], buildings (1.1m)[1,2], *airport*[1,2], power supply[2], roads[2], bridges[2], ports[2] | (Blanc and Strobl, 2016; Espada, 2018; Lagmay et al., 2015; Mas et al., 2015; NDRRMC, 2013; Soria et al., 2015) |
| Bopha (Pablo), 2012 | 1,060 | 1,022 | 1.04 | - | Wind[2], Flood[2] | Agriculture[2], buildings (217k)[2], power and water supply[2], and roads and bridges[2] | (NDRRMC, 2012) |
| Nesat (Pedring), 2011 | 172 | 437 | 0.39 | - | Wind[2], Flood[2], Surge[2], Slide[2,3] | Agriculture[2], buildings (44k)[2], power supply[2], dikes[2], roads and bridges[2] | (NDRRMC, 2011) |
| Megi (Juan), 2010 | 526 | 393 | 1.34 | - | Wind[2], Flood[2], Slide[2] | Agriculture[2], buildings (104k)[2], power supply[2], and roads and bridges[2] | (NDRRMC, 2010) |
| Parma (Pepeng), 2009 | 23 | 990 | 0.02 | - | Flood[1,3], Slides[1] | Agriculture[2], buildings (61k)[2], power supply[2], dikes[2], roads and bridges[2] | (Abon et al., 2011; Cooper and Falvey, 2009; Cruz and Narisma, 2016; Espada, 2018; Inokuchi et al., 2011; Nakasu et al., 2011; NDCC, 2009a, 2009b) |
| Ketsana (Ondoy), 2009 | 1 | 401 | 0.002 | - | Flood[1,3] *(Manila)*, Slides[3] | Agriculture[2], buildings (185k)[2], power supply[2], dikes[2], roads and bridges[2] | |
| Fengshen (Frank), 2008 | 9,286 | 465 | 19.96 | 41.2 | Flood[1,2,3], Surge[2], Slides[1,2] | Agriculture[2], buildings (407k)[2], power and water supply[2], and roads and bridges[2] | (Espada, 2018; IFRC, 2009; Yumul et al., 2012) |
| Xangsane (Milenyo), 2006 | 100,440 | 263 | 381.70 | 63.9 | Flood[1,3], Slides[1] | Dikes[1], power supply[1] | (Yumul et al., 2008) |
| Angela (Rosing), 1995 | 156,750 | 937 | 167.32 | 77.2 | Flood[2], Surge[2], Slides[2] | Agriculture[2], buildings (>96k)[2], power supply[2], dams[2], and roads and bridges[2] | (Joint Typhoon Warning Center, 1995) |

| | | | | | | | |
|---|---|---|---|---|---|---|---|
| Teresa (Katring), 1994 | 17,731 | 299 | 59.24 | 46.5 | Flood[3] | No report evaluated | |
| Flo (Kadiang), 1993 | 120 | 984 | 0.12 | - | Slide[3] | No report evaluated | |
| Ruth (Trining), 1991 | 95 | 564 | 0.17 | - | | No report evaluated | |
| Gordon (Goring), 1989 | 347 | 408 | 0.85 | - | | No report evaluated | |
| Dan (Saling), 1989 | 20,398 | 396 | 51.55 | 48.6 | | No report evaluated | |
| Skip (Yoning), 1988 | 182 | 1,120 | 0.16 | - | | No report evaluated | |
| Nina (Sisang), 1987 | 5,643 | 480 | 11.75 | 29.8 | | No report evaluated | (Espada, 2018) |
| Georgia (Ruping), 1986 | 2 | 343 | 0.01 | - | | No report evaluated | |
| Agnes (Undang), 1984 | 174 | 875 | 0.20 | - | | No report evaluated | |
| Irma (Anding), 1981 | 305 | 279 | 1.09 | 22.8 | | No report evaluated | |
| Betty (Aring), 1980 | 173 | 897 | 0.19 | 18.3 | Slide[3] | No report evaluated | (Espada, 2018) |

**Code availability and data availability**

The full array of fitted impact function parameters can be found as a Supplement of this paper. The scripts reproducing the main results of the paper and the figures are available at https://github.com/CLIMADA-project/climada_papers (Aznar-Siguan et al., 2020). The CLIMADA repository (Aznar-Siguan and Bresch, 2019; CLIMADA-Project, 2019) is openly available (https://github.com/CLIMADA-project/climada_python) under the GNU GPL license (GNU Operating System, 2007). The documentation is hosted on Read the Docs (https://climada-python.readthedocs.io/en/stable/), including a link to the interactive tutorial of CLIMADA. CLIMADA v1.4.1 was used for this publication, which is permanently available at the ETH Data Archive: http://doi.org/10.5905/ethz-1007-252 (Bresch et al., 2020).

**Author contribution**

SE and DB developed the idea and basic methodology of the paper. Coding and analysis were the work of SE and SL. SE prepared the manuscript with contributions from all co-authors.

**Competing interests**

The authors declare that they have no conflict of interest.

**Acknowledgements**

We would like to thank Gabriela Aznar-Siguan for her input regarding the platform CLIMADA and the implementation of the TC impact model in Python. Together with Benedikt Knüsel, she also supervised the Master Thesis of SL that contributed to the methods of this paper. Furthermore, our thanks go to Thomas Röösli, Benoît Guillod, and Boris Prahl, who helped shaping the calibration methodology, as well as Thomas Röösli, Maurice Skelton, Jamie McCaughey for their

substantial inputs during the internal review of the manuscript. Finally, we want to thank all members of the Weather and Climate Risks Group at ETH Zurich for their inputs and discussions shaping this publication. We would like to thank two anonymous referees for their thorough and valuable reviews.

**Financial support**

This research has been supported by the Innosuisse – Schweizerische Agentur für Innovationsförderung (grant no. 26792.1 PFES-ES).

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
