# Peer review of "Regional tropical cyclone impact functions for globally consistent risk assessments"

_Natural Hazards and Earth System Sciences, 2020_

## Referee Comment (RC1) · Anonymous Referee #1 · 4 Sep 2020

This manuscript by Eberenz et al. evaluates the model-simulated damages from tropical cyclones, and provides suggestions to improve this assessment and reduce the uncertainty of simulations by using regionally calibrated data. While the premise of the paper seems straightforward (e.g. "improving the calibration of the model will result in closer simulation of observed events"), the execution of the work in the paper is well done as it explores the limitations of their proposed approach. While overall the manuscript is well-presented and organised, there are opportunities to improve the text, particularly the analysis in the case study for the Philippines.

1) Some initial minor comments include better consistency in the risk language used in the paper; overall, it is good but there are some errors, e.g. para 35: "natural risk" –> is this hazard? risk? and para 180 "from natural catastrophes are records are available"

[Figure]

and following line ("natural disasters"). As you might well know it is a common refrain in the disaster/hazard community that "there are no natural disasters" – so do double check for the consistency. I also think that in para 55 where you say "one [...] function... might be inappropriate [...]", as this is the main argument of your work, could be made even stronger to create a deeper impression of the purpose of your research.

2) I find it interesting that the model over-simulates damages in the NWP basin, as it could be easily imaged that the damage function would not be able to simulate the myriad impacts of the associated hazards you mention such as the storm surges. In line with this I think it would have been interesting to provide an hypothesis for your research question, for example in para 70 where you pose this question.

My main comments are related to the case study of the Philippines.

4) Firstly, there is some confusion in the TC nomenclature which should be addressed for consistency (Table A4). For example, Ondoy is the local name and Ketsana is the international name.

5) Additionally this case study of the Philippines is very brief and only an assessment of asset exposure, and not vulnerability. I think that this section could use more context of the vulnerabilities associated with the Metro Manila region, enhanced by locally-led scientific literature on vulnerability (e.g. Porio 2011) as well as an analysis of the hazard events themselves to give the reader more context (see e.g. the work of Lagmay et al, Abon et al re: Ketsana and Haiyan (incl. effect of mountain ranges to improve your Done 2019 reference), Cayanan et al 2011 and Cruz/Narisma on SW monsoon effect on TCs, Yumul et al on TC Fengshen). Indeed I think these references could also be visited as Espada (2018) is often your only reference (Table A4).

5) There are some paragraphs that could use more attention and more geographical nuance, for example para 470 on Typhoon Haiyan (2013). This TC impacted mainly Tacloban City; indeed, this reflection of imbalanced damages appears to be simulated in your model output (Figure 6c) but Iloilo and Cebu cities are mentioned in lieu of this.

6) Having more context would also provide you an opportunity to refute an argument you pose earlier in your paper related to the CLIMADA setup, in that "[...] no impact is expected for low wind speeds," when it is evident many high-impact events in exposed and vulnerable regions cannot be estimated on wind speeds alone; geography/topography, exposure and vulnerability, local climate conditions (e.g. SW monsoon) play a significant and sometimes, larger role in realised damages from TCs.

7) With a better focus on this I think it would provide a richer and more meaningful assessment of exposure and vulnerability that give better context to your paper and the need for more regionalised calibration of damage estimates from TCs.

---

## Referee Comment (RC2) · Andrew Gettelman (Referee) · 14 Sep 2020

Review of Regional tropical cyclone impact functions for globally consistent assessments by Eberenz et al.

This manuscript dives into loss functions used in a Tropical Cyclone (TC) damage model and tries to adjust damage functions by region to better match the observed record of damages. The paper is generally well written and should be published in Natural Hazards and Earth System Sciences with minor revisions.

I have some specific comments below, but I would like to see a bit more explanation for some of the figures and analysis. Especially, some of the appendix (and the two figures) could be folded into the main text. Also, it's not clear whether the trend for

simulated damages is an over or under estimation of damages before calibration and whether this is due to strong or weak storms. Maybe this is in the figures, and could be mentioned in the discussion/conclusions.

Also, the analysis focuses on tuning v-half. What would happen if you either used or added v-thresh (the minimum wind speed for damages) as an adjustment parameter? Would that help? Why or why not? Can you test it?

In addition, US Damage is conveniently a function of wind speed for a specific reason: damage is often insured loss and that does not include flood. Can you comment on that?

Wouldn't it be wise to check the large scale data against 'small scale' engineering data based on different structure types? Or generally, why is the damage different, is it a physical reason (buildings are stronger or weaker than the US.) or a social reason: lower capital, less cost to rebuild, lower value? It would be nice to discuss this, in the conclusions if necessary.

Finally, would this be applicable to other models beyond CLIMADA? Why or why not?

Specific Comments:

Page 3, L81: Figure 1 is hard to follow. I suggest that perhaps each panel can be labeled a,b,c, etc, and then referred to in the text, rather than focusing on the section numbers. I had to read this 3 times to follow it, and only some of the panels are discussed.

Page 3, L90: figure 1: careful with the arrows. For example the arrows in the second row probably point the wrong way. You want 2.2.4 and 'simulated damage' to point to 2.3.2. Not. 2.3.1 pointing to them. This highlights the nomenclature problem with the figure.easier to label a,b,c, etc.

Page 3, L103: chosen resolution.

Page 3, L106: this description is a little confusing, and I think it is because you need to be clear about terminology. What is a hazard? What is exposure data? Maybe start from the concept of damage = exposed assets x damage ratio, and damage ratio is an impact function x hazard intensity. I think those are the correct terms.

Page 4, L145 : can be constrained

Page 5, L175: is v-thresh fixed? Seems like you just vary v-half, but I can see how v-thresh depends on building type. I.e wood v. Stone.

Page 7, L226: is the 20% difference significant? Or is the goal here to make sure the 58% number from climada matches the 76% value from observations?

Page 7, L234: what does the distribution of EDR look like un optimized? Can you plot it?

Page 7, L238: a plot of the TDR by country would be useful too.

Page 7, L245: is a data point a matched storm event? I.e 43 of the 376 events have damage in the USA and Canada?

Page 8, L281: what if you fit v-thresh instead or in addition? Might this help? Why or why not?

Page 9, L286:figure a1 and a2 should be part of the main text. Comment further please on the uncertainties. Is damage higher or lower? Where? What are the general issues?

Page 9, L291: higher or lower?

Page 9, L300: maybe it's just showing and figA1 and A2, but I in you should show one more step before figure 5. It's too hard to interpret what the optimization is doing and whether simulated damage is generally too high or too low.

Page 10, L324:is there a clearer way (eg fig A1) to show h overestimation?

Page 19, L591: can you speculate a little more on what next steps might be? Modify or add a flood risk component?

---

## Author Response (AR1)

**Authors' response**

**Research article**: "Regional tropical cyclone impact functions for globally consistent risk assessments" (Nat. Hazards Earth Syst. Sci. Discuss., https://doi.org/10.5194/nhess-2020-229; in review, submitted on 09 July 2020)
**Authors:** Samuel Eberenz, Samuel Lüthi, David N. Bresch

We thank anonymous referee #1 and Andrew Gettelman for their comments, which have improved the quality of the manuscript. The original comments from the referees are listed below directly followed by our responses in *blue and italic* and changes to the manuscript in **blue and bold**. The marked-up version of the revised manuscript attached below shows all proposed changes.
* * *
**Table of content**
* * *
**1) Response to comments by anonymous referee #1**

1.0) This manuscript by Eberenz et al. evaluates the model-simulated damages from trop- ical cyclones, and provides suggestions to improve this assessment and reduce the uncertainty of simulations by using regionally calibrated data. While the premise of the paper seems straightforward (e.g. "improving the calibration of the model will re- sult in closer simulation of observed events"), the execution of the work in the paper is well done as it explores the limitations of their proposed approach. While overall the manuscript is well-presented and organised, there are opportunities to improve the text, particularly the analysis in the case study for the Philippines.

1.1a) Some initial minor comments include better consistency in the risk language used in the paper; overall, it is good but there are some errors, e.g. para 35: "natural risk" –> is this hazard? risk? and para 180 "from natural catastrophes are records are available"

and following line ("natural disasters"). As you might well know it is a common refrain in the disaster/hazard community that "there are no natural disasters" – so do double check for the consistency.

*We would like to thank the referee for pointing out the inconsistencies of terminology. We have revised wording in the paragraphes mentioned by the referee. As for the lines 38f, we suggest replacing "natural risk" with "risk from natural hazard".*
*In lines 178f, we suggest restructuring both sentences to tackle the issue raised above and improve readability at the same time.*

*On a side note: Originally, the wording "natural and technological disaster" was taken from the EM-DAT data description: "EM-DAT distinguishes between two generic categories for disasters: natural and technological." ([https://www.emdat.be/explanatory-notes](https://www.emdat.be/explanatory-notes), last accessed 29/09/202). Still, we agree with the comment that this wording is not necessarily the most broadly accepted.*

*Suggested changes to the manuscript:*
*L. 38f:* " **Risk from natural hazards** is frequently modelled as a function of severity and occurrence frequency, [...]"
*L. 178f:* " **Reported damage estimates for disasters worldwide** are available from the International Disaster Database EM-DAT (Guha-Sapir, 2018). EM-DAT provides  data  **per event and country**, including disaster type and subtype, date of the event, and impact **estimates** ."

1.1b) I also think that in para 55 where you say "one [...] function... might be inappropriate [...]", as this is the main argument of your work, could be made even stronger to create a deeper impression of the purpose of your research.

*Indeed, this paragraph in the introduction is formulated more cautiously than what previous studies and our results indicate. Therefore, we follow the referee's suggestion to reformulate the sentence to express our position in a more confident way:*

*L. 56ff:*
"However, due to global heterogeneities in the tropical cyclone climatology (Schreck et al., 2014), building codes, and other socioeconomic vulnerability factors (Yamin et al., 2014), **it is inadequate to use a single**  universal impact function  for global TC risk assessments."

1.2) I find it interesting that the model over-simulates damages in the NWP basin, as it could be easily imaged that the damage function would not be able to simulate the myriad impacts of the associated hazards you mention such as the storm surges. In line with this I think it would have been interesting to provide an hypothesis for your research question, for example in para 70 where you pose this question.

*This point is well taken! An attribution of the interregional differences in vulnerability to TC damage to the drivers of vulnerability would be really interesting to look at. While such an attribution study would be out of scope of this study, we agree that it is important to discuss the different drivers and factors determining the vulnerability, that is, the shape of the calibrated impact functions.*

*In the manuscript, this is already touched upon, e.g. in the introduction:*
*L. 56f:*
*"[...] global heterogeneities in the tropical cyclone climatology (Schreck et al., 2014), building codes, and other socioeconomic vulnerability factors (Yamin et al., 2014) [...]"*

*… and in the discussion:*
*L. 501ff:*
*"[...] the results for the North West Pacific region (WP4), consisting of Japan, South Korea, Macao, Hongkong, and Taiwan, deviate substantially from GAR 2013. Simulated relative AAD in the region ranges from 0.2-0.8 ‰ as compared to 3.1 ‰ in GAR 2013. This difference implies that, besides the use of building type specific impact functions, the TC impact model of GAR 2013 substantially overestimates TC damages in WP4 compared to reported data."*
*In order to give a clearer message regarding attribution to the readers of the paper, we propose to add the following brief paragraphs to the manuscript:*

*In Section 1 (Introduction):*
**"While the attribution of vulnerability to regional drivers is outside the scope of this study, the results can serve as a starting point for further research disentangling the socio-economic and physical drivers determining vulnerability to TC impacts locally and across the globe."**

*In Section 6 (Conclusion and Outlook):*
**"The substantial over-estimation of TC damages in the North West Pacific with the default impact function opens the question for the drivers of the apparently lower vulnerability in this region. Considering the inability of the model setup to directly represent the impacts from TC surge and pluvial flooding, one would rather expect aggregated calibrated impact functions to be steeper than the default wind impact function. Therefore, we suggest investigating interregional differences in possible other drivers, including building standards but also damage reporting practices."**

My main comments are related to the case study of the Philippines.

1.3) Firstly, there is some confusion in the TC nomenclature which should be addressed for consistency (Table A4). For example, Ondoy is the local name and Ketsana is the international name.

*We would like to thank the referee for helping to clarify the confusion around the Typhoon names in the Philippines. Both in the text and in Table A4, we will adapt the following event names for consistency, mentioning the local names in brackets:*

- *"Bopha" →* **"Bopha (Pablo)"**

- *"Pedring (Nesat)" →* **"Nesat (Pedring)"**

- *"Pepeng (Parma)" →* **"Parma (Pepeng)"**

- *"Ondoy (Ketsana)" →* **"Ketsana (Ondoy)"** *(also in L. 459 and Figure 6c)*

- *"Fengshen (Franck)" →* **"Fengshen (Frank)"**

*Change in L. 422f, also removing double mentioning of water supply:*
*"* **TCs in the Philippines** *inflict damage on several sectors, most costly on housing and agriculture, but also on schools and hospitals, power and water supply, roads, and bridges (Table A4). Single events were also reported to damage and disrupt airports and ports (Typhoon Haiyan)* **and** *dikes (* **Nesat and Xangsane***"

1.4) Additionally this case study of the Philippines is very brief and only an assessment of asset exposure, and not vulnerability. I think that this section could use more context of the vulnerabilities associated with the Metro Manila region, enhanced by locally-led scientific literature on vulnerability (e.g. Porio 2011) as well as an analysis of the hazard events themselves to give the reader more context (see e.g. the work of Lagmay et al, Abon et al re: Ketsana and Haiyan (incl. effect of mountain ranges to improve your Done 2019 reference), Cayanan et al 2011 and Cruz/Narisma on SW monsoon effect on TCs, Yumul et al on TC Fengshen). Indeed I think these references could also be visited as Espada (2018) is often your only reference (Table A4).

*See combined reply to 1.4 and 1.5 in next point.*

1.5) There are some paragraphs that could use more attention and more geographical nuance, for example para 470 on Typhoon Haiyan (2013). This TC impacted mainly Tacloban City; indeed, this reflection of imbalanced damages appears to be simulated in your model output (Figure 6c) but Iloilo and Cebu cities are mentioned in lieu of this.

*Thank you very much for providing additional insights and references with regards to the case study of the Philippines. We answer both comments 1.4 and 1.5 together, since the improvements requested in 1.5 can be improved with reference to the additional literature suggested in comment 1.4.*

*We agree to the referee that integrating more local context and geographical accuracy will improve the informative content and accuracy of the case study. Doing so actually helps sharpen the argument of the whole of Section 4, that is, adding further support for the hypothesis that (1) a differentiation of urban and rural exposures and vulnerabilities would*

*increase the accuracy for sub-regional TC damage simulations, and (2) explicitly modeling of damage caused by sub-perils like storm surge and torrential rainfall would substantially improve TC damage simulations in the Philippines. Thus, an integration of the additional information will improve the manuscript while not affecting but rather strengthening the conclusions we draw from the case study.*

*At the same time, the case study is intended as an explorative excursion and not the main focus of the paper. Therefore, we suggest to incorporate the additional context most relevant for the discussion while not overly inflating the text body of Section 4 altogether.*

*As a consequence, we propose to incorporate additional local context provided by the proposed references in Sections 4.1 to 4.3 of the manuscript as follows:*

*Section 4.1 Tropical cyclones in the Philippines:*
*L. 419f:*
"In summary, TCs making landfall in the Philippines cause damage due to large wind speed, storm surge, as well as rain induced floods and landslides. **Meteorologically, the storm systems interact with the monsoon season, affecting both dynamics and the severity of torrential rain (Bagtasa, 2017; Cayanan et al., 2011; Yumul et al., 2012). TCs in the Philippines**  inflict damage on several sectors [...]"

*Section 4.2 Urban vs. rural exposure:*
*L. 430ff:*
"Most of the asset exposure value of the Philippines is concentrated around the metropolitan area of Manila **(Metro Manila).** **Located** around 14.5°N, 121.0°E (Fig. 7a)**, Metro Manila is Philippine's political and socio-economic center (Porio, 2011)**. The Typhoons Angela (1995), Xangsane (2006), and Rammasun (2014) are prominent TCs hitting **Metro Manila** directly. In our analysis,  these TCs come with particularly large EDRs, [...]"

*In Section 4.3 Impact of storm surge and torrential rain:*
*L. 461ff:*
"**While urban vulnerability to strong winds in Metro Manila appears to be overestimated by the calibrated impact function, Metro Manila is known to be highly exposed and vulnerable to regular, large scale flooding (Porio, 2011). The main drivers of flood vulnerability are its geographical setup, largely unregulated urban growth and sprawl, and substandard sewerage systems, especially in low-income areas (Porio, 2011).** Tropical Storm **Ketsana, locally known as** Ondoy (2009) is an example with very low simulated damages **coinciding with large reported damages associated to flood in Metro Manila: Ketsana's**  EDR is 0.002, i.e. its simulated damage is more than two orders of magnitude smaller than reported. The large reported damage (NRD=401 million USD) was mainly due to floods and landslides: Torrential rainfall caused severe river flooding in the **Metro Manila** and landslides around Baguio City, resulting in severe damages (Abon et al., 2011; Cruz and Narisma, 2016; Nakasu et al., 2011; NDCC, 2009a).

 **The flood** damages were not resolved by the wind-based impact model, with intensities well below 50 ms-1 and neither affecting Manila nor the northern Baguio City directly (Fig. **7**d). **Notably, even for TCs with large overestimation of simulated damage due to high wind speeds in Metro Manila, namely Fengshen and Xangsane, a substantial part of the reported damage was actually caused by pluvial flooding and landslides and not by wind alone (Yumul et al., 2008, 2011, 2012).**"

L. 470ff:
"It should be noted that  sector specific impacts are not resolved in the impact model and Haiyan did not affect Manila directly. **R**elatively large damages were simulated **around Tacloban City, Leyte, which was actually devastated by Haiyan's storm surge. Large wind impacts were also simulated further West** around the cities Iloilo and Cebu (Fig. **7**c) **that were not as exposed to surge as Leyte province.**"

*In Table A4:*
*Add references per event, associated disasters, and affected assets for TCs Haiyan, Ketsana, Fengshen, and Xangsane. Additional information are based on the following publications:* **Abon et al. (2011), Cruz and Narisma (2016), Lagmay et al. (2015), and Yumul et al. (2008, 2011, 2012)**.

1.6) Having more context would also provide you an opportunity to refute an argument you pose earlier in your paper related to the CLIMADA setup, in that "[...] no impact is expected for low wind speeds," when it is evident many high-impact events in exposed and vulnerable regions cannot be estimated on wind speeds alone; geography/topography, exposure and vulnerability, local climate conditions (e.g. SW monsoon) play a significant and sometimes, larger role in realised damages from TCs.

*See point 1.2 above and combined reply to 1.6 and 1.7 in next point.*

1.7) With a better focus on this I think it would provide a richer and more meaningful assessment of exposure and vulnerability that give better context to your paper and the need for more regionalised calibration of damage estimates from TCs.

*Reply to both comments 1.6 and 1.7: We agree that the limitations of a TC impact model based on wind speed as the only hazard intensity can not be stressed enough. Generally, we believe that the manuscript reflects upon these limitations and uncertainties to a sufficient degree and the need for more regionalised calibration, especially for local applications, have already been firmly emphasized in the discussion and outlook sections of the manuscript. For instance, in the last paragraph of Section 6:*
*"Limitations of our research motivate future work. For TC impact models, we echo the call for a more refined representation of TC hazard as a combination of wind, surge, and rain induced flood and landslide events. Furthermore, our case study for the Philippines suggests that differentiating between urban and rural asset exposure, considering topography in wind speed*

*estimations, and the inclusion of exposed agricultural assets could further increase model accuracy." (L. 587f)*

*The case study on the Philippines, with the improvements suggested in comment 1.4 and 1.5, adds detail and evidence of the discussion of these uncertainties and recommendations for further model development in the outlook.*

*Regarding the specific argument regarding low wind speeds from Section 2.2.3, we suggest to narrow the statement and also emphasise the learnings from the case study in these regards in Section 4.4, see suggested changes in the manuscript below:*

*L. 144 (Section 2.2.3):*
"Since no  **directly wind induced damage** is expected for low wind speeds, [...]"

*L. 476ff (Section 4.4):*
"The case of the Philippines reveals limitations of the model and calibration due to the lack of an explicit representation of sub-perils such as storm surge, torrential rainfall, and landslides (Sect. 4.3). **The flood damage caused by Ketsana is a showcase example for severe damages associated with a TC with relatively low wind speeds, that is, an event that cannot be adequately reproduced with a wind-based impact function.** Adding to the stochastic uncertainty, the magnitude of rainfall during a TC events in the Philippines is not only determined by the intensity of the TC event, but also by the coinciding monsoon season, as in the case of the Typhoons Fengshen and Haiyan (Espada, 2018; IFRC, 2009**; Yumul et al., 2012**)."
* * *
**2) RESPONSE TO COMMENTS BY REFEREE #2 (ANDREW GETTELMAN)**

2.0) Review of Regional tropical cyclone impact functions for globally consistent risk assessments by Eberenz et al.
This manuscript dives into loss functions used in a Tropical Cyclone (TC) damage model and tries to adjust damage functions by region to better match the observed record of damages. The paper is generally well written and should be published in Natural Hazards and Earth System Sciences with minor revisions.

2.1) I have some specific comments below, but I would like to see a bit more explanation for some of the figures and analysis. Especially, some of the appendix (and the two figures) could be folded into the main text.

*Thank you for the suggestions regarding the figures and explanations in the manuscript. They are most welcome and we will reply more in-depth and suggest changes together in our answers to the specific comments below.*

*Most notably, instead of folding figures A1 and A2 into the main text, we suggest a new figure (EDR boxplot per region, c.f. Comment 2.13) to be added to the results section, c.f. responses to comments 2.13, 2.17, 2.19, and 2.20.*

2.2) Also, it's not clear whether the trend for simulated damages is an over or under estimation of damages before calibration and whether this is due to strong or weak storms. Maybe this is in the figures, and could be mentioned in the discussion/conclusions.

*Whether damages are on average over- or underestimated is shown explicitly in the manuscript: For the average per region, Figure 5 and Table A2a provide this information as conveyed by uncalibrated total damage ratio (TDR), i.e., the grey bars in Figure 5b, next to the values after calibration. These numbers are also discussed in the text (L. 288ff, Section 3.1.1). For the single countries the spread of uncalibrated event damage ratios (EDR) information is displayed in Figure S2 in the supplementary materials. We agree that EDR per region could be shown more explicitly in the main text. The regional findings with regards to over- and underestimation before calibration form the basis for the whole calibration and are reflected both in the results and discussion section.*

*As for the question whether strong or weak storms are driving the average over- or underestimation (i.e. as measured by TDR), we agree to the referee that this is not yet discussed broadly in the manuscript. Our results show no significant correlation between normalized reported damage (NRD, here taken as a measure of TC severity) and EDR (measure of over-/underestimation of damage in CLIMADA). In response to this comment, we also plotted scatter plots of EDR vs NDR, finding no evidence of any significant relationship between TC severity and over- or underestimation.*

*We will add these scatter plots to the appendix of the manuscript as new* **Figures A3***:*

[Figure]

**"Figure A3: No significant correlation between event damage ratio (EDR) and normalized reported damage (NRD) was found. The scatter plots show the relationship for 473 TC events worldwide computed with three different sets of impact functions: (a) uncalibrated default ($V_{half}$=74.7 ms-1), (b) RMSF optimized, and (c) TDR optimized. The nine calibration regions are differentiated by colour."**

*Suggested addition to Section 3.2.2.:*
*L. 298: "The EDR values within regions show a large spread over several orders of magnitudes (Fig A1).* **There is no significant correlation between EDR and NRD (Fig. A3), suggesting that the over- and underestimation of simulated event damages is not related to TC severity.** *The largest spread, as expressed by the RMSF [...]"*

*To clarify that Figure 5b shows over/underestimation of average damages per region, we will explain this better in the figure caption of Figure 5:*

*L. 337ff (caption Figure 5):*
*"Figure 5: Calibration results and cost functions for nine calibration regions and all regions combined,* **each shown before (grey) and after calibration (blue and red)**: *(a) Vhalf: fitted impact function parameter; (b) TDR: ratio of total simulated and normalized reported damage; (c) RMSF: root-mean-squared fraction; and (d) AAD:* **normalized reported (green) and simulated** *annual expected damage* **(AAD)**. *[...]"*

*Furthermore, in our response to comment 2.13, we suggest to show uncalibrated EDR per region in a new figure in the beginning of the results section.*

2.3) Also, the analysis focuses on tuning v-half. What would happen if you either used or added v-thresh (the minimum wind speed for damages) as an adjustment parameter? Would that help? Why or why not? Can you test it?

*This comment touches upon one of the most pivotal decisions during calibration: the choice of free parameters in the impact function. Thank you for pointing out that more justification for the decision to only vary $V_{half}$ should be provided. The very short reason is that we concluded that fitting more than one of the linear dependent parameters (c.f. L. 154f) increases the risk of overfitting. Here is the long justification.*

*The approach of this paper builds on the Master Thesis by one of the authors with the title "Applying Machine Learning Methods to the Assessment of Tropical Cyclone Impacts" (Samuel Lüthi, 2019, available at https://doi.org/10.3929/ethz-b-000398592).*
*In the Master Thesis, regional TC impact functions were calibrated with Bayesian optimization methods based on almost the same data set of TC tracks and EM-DAT entries per event and country. Also, the same wind field model and impact engine as implemented in CLIMADA was applied. Differences between the Lüthi (2019) and the present paper under discussion mainly lay in the definition of the regions but also further refinements and quality control of the underlying data added for the present paper: EMDAT events which produce no damage in CLIMADA had not been excluded from the analysis in Lüthi (2019). This can cause artefacts, e.g. if a Carribean TC undergoes extratropical transition and then makes "landfall" a second time as a low pressure system.*

*Lüthi compared regional calibration results of two multi-parameter impact functions: (i) a sigmoid function similar to the function by Emanuel (2012) with three free parameters (slope, offset, and maximum intensity) and (ii) a 12-step staircase function. Figures 3.4 and C.1 of Lüthi (2019) show the resulting impact functions and RMSF scores:*

[Figure]

FIGURE C.1: Estimated vs. actual damages for all storms of all regions (left, log-scale) using sigmoid damage functions, including regional damage functions (top-right), and regional training (left box per region) and validation (right box per region) RMSF error (bottom-right). Damages are calculated using the regional damage functions which result from the calibration using all data per region. Training and validation RMSF are the output of a 10-fold cross validation per region. This plot corresponds to Figure 3.4.

[Figure]

FIGURE 3.4: Estimated vs. actual damages for all storms of all regions (left, log-scale), including regional damage functions (top-right), and regional training (left box per region) and validation (right box per region) RMSF error (bottom-right). The solid line indicates correctly estimated storms, storms in between the two dotted lines are estimated in the right order of magnitude. Damages are calculated using the regional damage functions which result from the calibration using all data per region. Training and validation RMSF are the output of a 10-fold cross validation per region.

*Based on his results as partially shown in the two figures displayed above, Lüthi (2019) concludes: "The comparison of different calibration approaches (Section 3.3) reveals that sigmoid functions are an attractive tool. As these functions depend solely on three parameters, they can be calibrated comparatively fast and at lower computational cost. However, they can produce counter-intuitive results. As an example, the resulting damage function for East Africa (Figure C.1, appendix) shows mean damage degree values larger than zero at zero wind. On the other hand, the multi-step function shows a similar performance using twelve parameters." (p. 25) and "While the damage functions look quite different, the estimated damages and also the training and validation errors are similar. This is due to the fact that the damage functions are quite similar in the region of 30-50 m/s." (p. 41)*

*For the parameterization used in the present study, this implies that a shift of $V_{tresh}$ alone would require relatively large shifts to unrealistically low or even negative wind speeds (Indian Ocean) or very large intensities (North West Pacific region). Changing multiple parameters is problematic because the parameters $V_{thresh}$, $V_{half}$, and **scale** are not linear independent. This introduces a risk of overfitting considering the large uncertainties in the damage data underlying the calibration; Fitting a more flexible, multi-step function with larger resolution from 30-50m/s could be a valid alternative to the sigmoidal function. However, as Lüthi (2019) showed, this more computationally expensive approach does not result in any considerable improvement in skill. Therefore, we decided to keep the fitting as simple and transparent as possible as long as the other uncertainties are not reduced.*

*In light of the findings by Lüthi (2019), our approach comes with the following advantages: (i) thanks to the fixed value of $V_{thresh}$, it only produces physically plausible curves, (ii) allowing only 1 free parameter yields a relatively fast (computationally cheap) calibration and (iii) results are easy to interpret and compare. Comparing the RMSF values after calibration of the two studies shows that the results are generally similar; often RMSF numbers retrieved for this study are lower. In the case of China (WP3), the only region which has the same underlying countries in both studies, the calibration in this study produces lower RMSF values. It should be noted that RMSF in Lüthi (2019) are negatively affected by events which produce no damage in CLIMADA (as mentioned above, these events have been excluded in the analysis for this paper).*

*In response to this comment and also comment 2.11 below, we suggest to add a brief discussions of the choice of fitting parameters with reference to Lüthi (2019) in the manuscript, that is, in Section **2.2.3 Impact Function**, L. 165ff:*

"**In a comparison of calibration results based on a sigmoidal impact function with a more complex 12-step staircase function, Lüthi (2019) found no improvement of calibration skill with the more complex function. Therefore, we use the sigmoidal function in this study.** We define a default impact function with $V_{thresh}$ = 25.7 ms$^{-1}$ and $V_{half}$ = 74.7 ms$^{-1}$ that is used for a first, uncalibrated, simulation of global TC damages, and as a starting point for calibration. While $V_{half}$ is fitted during the calibration process, we keep the lower threshold $V_{thresh}$ constant throughout the study. **This is based on the finding by Lüthi (2019) that the variation of more than one of the linearly dependent parameters most likely results in an overfitting during calibration, with physically implausible values for $V_{thresh}$ in some world regions.**"

2.4) In addition, US Damage is conveniently a function of wind speed for a specific reason: damage is often insured loss and that does not include flood. Can you comment on that?

*Your point is well taken and we thought about this before starting our work. As for flood-related (as well as for rain and surge) damage, that's a wider field. Indeed standard US policies do not cover flood (or explicitly state an exclusion). But in recent cases (such as Katrina, 2005), the insurance commissioner forced direct insurance companies to cover damages 'as they occurred' and many (if not most) times even loss adjusters could not separate (wind/surge/rain) in hindsight, hence reported damages do often include the flooding component. But as the aim of the present effort is to provide a globally consistent and readily available impact model, calibrating to EM-DAT provides such a global yet at least regionally adjusted perspective - and it is not quite clear what EM-DAT reported damages cover in detail (some numbers look rather like total direct rather than only insured damage, some event reportings might even include total economic damage, inclusive - some - business interruption, i.e. indirect impacts). That's why we consider EM-DAT as a lower bound and rather 'best guidance' than 'ground truth' (which unfortunately can not be established in hindsight).*

2.5) Wouldn't it be wise to check the large scale data against 'small scale' engineering data based on different structure types? Or generally, why is the damage different, is it a physical reason (buildings are stronger or weaker than the US.) or a social reason: lower capital, less cost to rebuild, lower value? It would be nice to discuss this, in the conclusions if necessary.

*A comparison of the large scale calibration with socio-economic indicators and engineering based impact functions would certainly be a great gain for further research and improvement of the vulnerability component of TC impact models. However, this is a rather wide and complex field: The regional impact functions are a proxy for the aggregate of multiple types of damages to all kinds of different building types. At the same time, the quality of impact data is not good enough to differentiate between these. Furthermore, engineering data is often not publicly available and if so, only available for richer countries and building codes are hard to compare across regions.*

*A comparison against specific bottom-up data is beyond the scope of this paper that focussed on the question how much can be achieved with an event-based, top-down fitting of impact function against reported damage data.*
*The approach followed for GAR 2013 (Yamin et al., 2014) is definitely a good starting point to regionalize impact functions based on vulnerability indicators rather than empirically. However, our study shows the limitations of such an approach when no comparison to reported damages is done.*

*Following your suggestion, a welcome next step could be a study combining the empirical evidence provided by reported damage data on the one hand with building codes / socio-economic indicators on the other hand. This approach would be quite challenging, as it adds even more layers of complexity and cascading uncertainties to the calibration. Still, we agree that it could help to gain a better understanding of the drivers of the inter-regional differences in TC vulnerability. Sensitivity analysis on a more local and regional level could be a feasible starting point.*

*As comment 1.2 by referee #1 points into a similar direction (asking for hypotheses on the reasons for the inter-regional differences between calibrated impact functions), part of the changes to the manuscript we suggest here to take up are duplicates of our suggestions in reaction to comment 1.2. Please refer to AR1 for the full answer to 1.2.*

*Proposed changes in the manuscript:*

*In Section 1 (Introduction):*
**"While the attribution of vulnerability to regional drivers is outside the scope of this study, the results can serve as a starting point for further research dissecting the socio-economic and physical drivers and factors determining vulnerability to TC impacts locally and across the globe."**

*In Section 6 (Conclusion and Outlook):*

"**The substantial over-estimation of TC damages in the North West Pacific with the default impact function opens the question for the drivers of the apparently lower vulnerability in this region. Considering the inability of the model setup to directly represent the impacts from TC surge and pluvial flooding, one would rather expect aggregated calibrated impact functions to be steeper than the default wind impact function. Therefore, we suggest investigating interregional differences in possible other drivers, including protection and construction quality and standards – but also damage reporting practices. A study combining the empirical evidence provided by reported damage data on the one hand with socio-economic indicators on the other hand would be desirable but rather challenging, as this would add even more layers of complexity and cascading uncertainties to the calibration, especially on a global level.**"

2.6) Finally, would this be applicable to other models beyond CLIMADA? Why or why not?

*Generally yes, the impact functions are not specific to the CLIMADA modeling framework. Also, we would expect the relative inter-regional differences to be robust with other TC impact modeling set-ups, i.e. that the North West Pacific shows the lowest vulnerability as expressed by the flat impact function.*
*The precise shape and scaling of the calibrated impact functions are, however, to a certain degree specific to the decisions and components of the modeling setup, including most prominently:*

1. *The representation of hazard by wind alone: an explicit representation of surge and rain would require different impact functions, also for wind.*
2. *The choice of free parameters in the impact function, as already discussed in response to your comment 2.3.*
3. *The value of total asset values (TAV, c.f. Table A3): impact functions would scale differently with a different assumed total inventory value of exposed assets.*
4. *Spatial resolution: The impact functions are calibrated for 10 km resolution. Parameters could change, if hazard and exposure is represented on a higher or lower resolution.*

*The first point is now already reflected upon in the manuscript, most prominently in the discussion:*
*"The regionalized impact functions presented here were calibrated for wind-based damage modelling on a spatially aggregated level. Model setups with an explicit representation of related sub-perils like storm surge or torrential rain require different (i.e. flatter) impact functions for the wind-induced share of TC damage, as well as additional impact functions for all sub-perils. Likewise, impact models with an explicit representation of building types and agricultural assets require a more differentiated set of impact functions." (L. 562ff)*

*However, we agree that points 2, 3 and 4 could be stated more explicitly, to support the use of the study results outside CLIMADA. Therefore, we will add the following limitations to the discussion:*

*L. 562ff (Section 5.2 Uncertainties and limitations):*

"**While the results of this study are not specific to the CLIMADA modeling framework, the precise shape and scaling of the calibrated impact functions are, however, to a certain degree specific to the choices and input data of the modeling setup: (1) The choice of free parameters in the impact function (c.f. Section 2.2.3 and Lüthi, 2019); (2) The TAVs (c.f. Table A3): impact functions would scale differently with a different assumed inventory of exposed assets; (3) spatial resolution; and (4) the representation of hazard intensity:** The regionalized impact functions presented here were calibrated for wind-based damage modelling on a spatially aggregated level. [...]"

*Specific Comments:*

2.7a) Page 3, L81: Figure 1 is hard to follow. I suggest that perhaps each panel can be labeled a,b,c, etc, and then referred to in the text, rather than focusing on the section numbers. I had to read this 3 times to follow it, and only some of the panels are discussed.

2.7b) Page 3, L90: figure 1: careful with the arrows. For example the arrows in the second row probably point the wrong way. You want 2.2.4 and 'simulated damage' to point to 2.3.2. Not 2.3.1 pointing to them. This highlights the nomenclature problem with the figure.easier to label a,b,c, etc.

*Thank you for the well thought out suggestions to make Figure 1 easier to follow. We are taking up both suggestions and propose the following adjustments to Figure 1 and its caption and reference in the manuscript to improve readability:*

*L. 86:*
*Figure 1: (1) replace section numbers in panels by labels (a) to (h); (2) remove arrow heads pointing towards panels (e) and (d).*

[Figure]

*L. 89 (caption Figure 1):*
"Figure 1: Schematic overview of the data and methods applied to calibrate regional TC impact functions in a globally consistent manner. From left to right: TC event damages are first simulated within the CLIMADA framework based on TC  **hazard (a)**, asset exposure **(b)**, and a default impact function **(c), c.f. Sect. 2.1 to 2.2.3** . Resulting simulated damages **(d)** are  compared to reported damage data from EM-DAT **(e) for 473 matched TC events (f) by means of the damage ratio (g), c.f. Sect. 2.2.4 to 2.3.2** . During calibration **(h), steps (c) to (g) are repeated several times with varied impact functions for each region, optimizing the cost functions TDR and RMSF (c.f. Sect. 2.3.3)** . The result is a set of best fitting impact functions for nine world regions **(Sect. 3.2)**. Finally, the calibrated impact functions are plugged into CLIMADA **once more** (d**ashed arrow**) to compute annual average damage per region **(Sect. 3.3)**."

*L. 80ff:*
"To regionally calibrate TC impact functions, simulated damages are compared to reported damages, as illustrated in Figure 1:  In a first step, direct economic damage caused by TCs are simulated in the impact modelling framework CLIMADA (**Fig. 1a-d,** Sect. 2.1 to 2.2.2) with one single default impact function applied globally to start from (Sect. 2.2.3). Then, damage data points per country and storm are assigned to entries of reported damage (**Fig. 1e-f,** Sect. 2.3.1). For the matched events, the ratio between simulated and reported damage is calculated (**Fig. 1g**, Sect. 2.3.2). For calibration, countries are clustered into regions and two complementary cost functions are optimized based on the damage ratios, by regionally fitting the slope of the impact function (**Fig. 1h**, Sect. 2.3.3)."

2.8) Page 3, L103: chosen resolution.

*Suggested change in L.103:*
"The setup  work**s** equally well at higher **chosen** resolution [...]"

2.9) Page 3, L106: this description is a little confusing, and I think it is because you need to be clear about terminology. What is a hazard? What is exposure data? Maybe start from the concept of damage = exposed assets x damage ratio, and damage ratio is an impact function x hazard intensity. I think those are the correct terms.

*Suggested change in L.103ff:*
"**In the CLIMADA framework, damage is defined as the product of  exposed assets and a damage ratio. In our case,**  damage per TC event and country is  **simulated** as  **follows: For each grid cell and event, damage is calculated as the product of total exposed asset values and the mean damage ratio. The mean damage ratio (0 to 100%) results from plugging the hazard intensity (maximum sustained wind speed) into the impact function. Finally, damage per event is aggregated over all grid cells within the country.**

2.10) Page 4, L145 : can be constrained

*We correct the typo in L. 145 ("*constraint*" to "***constrained***").*

2.11) Page 5, L175: is v-thresh fixed? Seems like you just vary v-half, but I can see how v-thresh depends on building type. I.e wood v. Stone.

*As already mentioned in response to comment 2.5, the regional impact functions are a proxy for
the aggregate of multiple types of damages to all kinds of different building types. Due to the
global and aggregated scope of this study, our approach did not start from differences of
$v\_thresh$ with regards to building types but rather explored how far we get with a top-down
approach that is "blind" to bottom-up specifications.*
*Still, we agree that a variation of $v\_thres$ could be worthwhile. We have explained our reasons
to vary $v\_half$ only in the answer to comment 2.3 above. Please refer there for the detailed
response and also the additional clarifications with reference to Lüthi (2019) we suggest to add
the manuscript before publication.*

2.12) Page 7, L226: is the 20% difference significant? Or is the goal here to make sure the 58%
number from climada matches the 76% value from observations?

*In Section 2.3.1, we state percentages to examine and illustrate what share of simulated and
reported TC damages is represented by the matched TC events, that is, considered in the
analysis. The fact that both normalized reported damages and simulated damages represent
more than half of the total damage inventory of the two data sets gives us confidence that a
representative calibration can be based on the matched events. If the shares were much lower,
we would be much less confident that the calibration results are representative.*

*In light of the reason to calculate these shares, the difference of 20% has not been of major
interest or concern to us: it simply reflects that the two datasets are both not necessarily
complete inventories of damaging TCs, and also that the reported damage data comes with
substantial uncertainties as discussed in Section 5.2. The mismatch between the event
inventories of the two data sets was one of the reasons to limit calibration on those events with
a validated match between the TC track from IBTrACs and the data point from EM-DAT.*

2.13) Page 7, L234: what does the distribution of EDR look like un optimized? Can you plot it?

*The distribution of un-optimized EDR can indeed be shown in a more concise fashion. The
distribution of uncalibrated EDR per calibration region is shown below as boxplots, highlighting
the differences between the regions. Furthermore, we plotted histograms for global EDR in
response to the referee's request: The histograms of the global distribution of uncalibrated EDR*

*show a large spread, as already discussed in the manuscript (L. 298ff), with both over- and underestimation of simulated damages occurring. Calibration reduces the spread to a certain degree, placing more than half of events in the EDR range from 0.1 to 10, that is, simulated event damage is of the same order of magnitude as normalized reported damage.*

[Figure]

*Figure: Spread of event damage ratio (EDR, boxplot) and total damage ratio (TDR) per region before calibration ($V_{half}$=74.7 ms$^{-1}$) per region. The plots are based on data from 473 TC events affecting 53 countries. The EDR boxplots show the median (green line), the first and third quartiles (IQR, blue box), data points outside the IQR but not more than 1.5·IQR distance from either the first or the third quartile (black whiskers), and outliers (black circles). The additional markers show TDR before calibration (green diamond).*

*For the distribution of EDR per country please refer to the boxplots in the Supplement (URL, updated figures also attached to the end of this document).*
*Change to the manuscript: We suggest to add the regional boxplots of EDR to the results Section 3.1 (L. 283, "Damage ratio with default impact function") the histogram to the supplement of the manuscript.*

2.14) Page 7, L238: a plot of the TDR by country would be useful too.

*We have already provided plots of the distribution of uncalibrated EDR per country in Figure S2 in the supplement. To satisfy your request for TDR per country, we added TDR (which can be read as a weighted average of EDR) to these plots, both before and after calibration.*

*The updated Figures S2a-f are attached below, at the end of this document, for your consideration.*

*Change in Section 2.3.2 of the manuscript:*
*L. 238f:*
**"The distribution of EDR and TDR before calibration as well as TDR after calibration is shown per region in Figures 6 and S4 and per country in the supplementary Figure S2."**

2.15) Page 7, L245: is a data point a matched storm event? I.e 43 of the 376 events have damage in the USA and Canada?

*Yes, the number of matched events per region is listed in Table A2.*
*To clarify L.245, we suggest the following change in the manuscript:*

*L.245:*
"a minimum desired number of 30 data points **(matched TC events)** per region"

2.16) Page 8, L281: what if you fit v-thresh instead or in addition? Might this help? Why or why not?

*To avoid redundancy between the responses, please refer to the replies to comments 2.3 and 2.11 for a discussion of the choice of free parameters in the impact function.*

2.17) Page 9, L286:figure a1 and a2 should be part of the main text. Comment further please on the uncertainties. Is damage higher or lower? Where? What are the general issues?

*The referee has a valid point in suggesting to add a figure showing the spread of EDR to the main text. In our opinion, it is not necessarily required to add Figures A1 and A2 to the main text to answer the questions asked with this comment, not to overload the manuscript with figures. The question whether and where damage is higher or lower before and after calibration is now already answered on an aggregated and more digestible level in Figure 5 within the main manuscript: Figure 5b compares TDR with and without calibration for each region and globally, Figure 5d shows normalized reported damage and total simulated damage per region, again with and without calibration. As for the spread of EDR (damage ratio of single events), we agree that this should be shown in the main text. We suggest showing this in the form of boxplots per region, as they also show the spread of EDR to get a better feeling for the uncertainties. The suggested Figure is shown in response to comment 2.13.*

*Ad uncertainties: The uncertainties are commented on quite extensively in Section 5.6 ("Uncertainties and limitations") already. The spread of uncalibrated EDR within each region shown in Figure A1 now already illustrates the quantitative extent of the uncertainties, this will be further improved with the suggested figure. Furthermore, the discrepancy between TDR and RMSF calibration gives a hint on how robust the calibration is for each region (Figures 5b and 5c). This is already discussed in the manuscript already, i.e. in line 319: "The comparison of complementary calibration approaches gives an indication of the robustness of the calibration per region." and line 520f: "The deviation between the results of the two calibration approaches indicates how robust the calibration is with regards to the model's ability to represent the correct*

*order of magnitude of single event damage. Whereas the model setup returns reasonable risk estimates and consistent calibration results for Central and North America, we found an extensive spread in EDR and calibration results for other regions, especially in East Asia."*

*Suggested changes to the manuscript: c.f. Comment 2.13.*

2.18) Page 9, L291: higher or lower?

*There are both cases, regions with higher and with lower simulated damages. As described in lines 291ff, uncalibrated TDR is below 1 in some regions and above 1 in others, c.f. answer to comment 2.2 above.*

*To clarify, we suggest the following reformulation in Section 3.2.1:*
*L. 289ff:*
"Both **the ratios EDR and the cost functions** RMSF  and TDR  **show inter-regional differences**,  **with regard to** the deviation of the damages simulated with the default impact function from reported damages **(Fig. 4 and 6)**, . **For most regions, total simulated and normalized reported damage deviates less than one order of magnitude (Table A2)** ."

2.19) Page 9, L300: maybe it's just showing and figA1 and A2, but I in you should show one more step before figure 5. It's too hard to interpret what the optimization is doing and whether simulated damage is generally too high or too low.

*This is a valid point: insights into the optimization and its consequences for damage ratios is a crucial aspect of this publication. As already mentioned in our responses to comments 2.2 and 2.19, Figure 5b does indeed show where simulated damages are generally too high or too low as compared to reported damages, both before and after calibration.*
*As for your request to get more insight in what the optimization is doing on a more detailed level, we hope that our responses and propositions in response to comments 2.2, 2.13, and 2.14 allow for a more thorough interpretation of what the implications of the calibration on a more detailed level, e.g. with regards to TC severity and per country. In our opinion, the additional figure suggested in response to comment 2.13 serves to show the additional step before Figure 5 as requested by the referee.*

*A further, more complex insight into the optimization on the basis of fitted $V_{half}$ is provided in Figure S3 in the supplement, providing insight in the robustness of TDR and RMSF to changes in the impact function. We agree that this could be mentioned more explicitly in the text, though, as proposed below.*

*Proposed changes in the manuscript:*
*L. 316f:* " **The sensitivity** of TDR and RMSF per region  **to changes in** Vhalf  **is visualized** in the Supplement**: Regions with a large uncertainty, i.e. a**

**large spread of EDR, generally show a relatively low robustness of the cost functions** (Fig. S3). **On a globally aggregated level, calibration reduces the spread of EDR to a certain degree, placing more than half of events in the EDR range from $10^{-1}$ to 10."**

2.20) Page 10, L324:is there a clearer way (eg fig A1) to show h overestimation?

*The over- and underestimation of TC damages as expressed by TDR (aggregated level) and EDR (single events) is shown in Figure 5 and Figures A1 and A2. For the aggregated level, TDR before and after calibration shows the average under- and overestimation aggregated per region in Figure (5b), c.f. Also our response to comment 2.2.*
*For the event level, please refer to the EDR plots per country as shown in Figure S2 in the supplement. Beyond this, we hope that our explanations and additional figures proposed in response to your comments 2.2 and 2.13 help to clarify the communication of over- and underestimation of TC damages.*

2.21) Page 19, L591: can you speculate a little more on what next steps might be? Modify or add a flood risk component?

*We would strongly suggest to work towards future TC risk assessments based on modelled TC events and an explicit representation of surge and rain.*

*Adding a storm surge component, requires high resolution to resolve topography.*

*Representation of torrential rain, requires to take into account transition speed of the TC, as in the case of Hurricane Harvey that stayed stationary over Houston for a long time, dumping more rain than expected in the same area. Also, interaction with other weather phenomena like monsoon need to be taken into account, as in the case of the Philippines.*

*To further improve the impact functions, it would be worthwhile to combine damage data based calibration with socio-economic and engineering type data, as discussed in response to comment 2.5. Especially if impact functions for different sub-perils (wind, surge, rain) need to be combined, more information and knowledge than the one provided by reported damage data is required to constrain calibration. This could also involve expert judgement and engineering based impact functions.*

*The last point can be stressed more in the manuscript, we therefore suggest to add the following sentences in the outlook (Section 6):*

*L. 589ff:*
**"When modeling multiple TC sub-perils, aggregated reported damage data are not sufficient to constrain impact function calibration. This might be resolved by consulting socio-economic and engineering type data and knowledge."**

[Figure]

*Appendix to AC2: Reworked Figures from the Supplement (Figures S2, c.f. comment 2.14*
*above):*

**Figure S2a: Spread of event damage ratio (EDR, uncalibrated) and total damage ratio (TDR) per country in the North Atlantic and North East Pacific basin (NA). The plots are based on data from 23 countries. The EDR boxplots show the median (green line), the first and third quartiles (IQR, blue box), data points outside the IQR but not more than 1.5·IQR distance from either the first or the third quartile (black whiskers), and outliers (black circles). The additional markers show TDR before calibrated (green diamond) and after calibration (blue circle: RMSF optimized and red squares: TDR optimized).**

[Figure]

**Figure S2b: Spread of event damage ratio (EDR, uncalibrated) and total damage ratio (TDR) per country in the North Indian Ocean basin (NI). The plots are based on data from six countries. The EDR boxplots show the median (green line), the first and third quartiles (IQR, blue box), data points outside the IQR but not more than 1.5·IQR distance from either the first or the third quartile (black whiskers), and outliers (black circles). The additional markers show TDR before calibrated (green diamond) and after calibration (blue circle: RMSF optimized and red squares: TDR optimized).**

[Figure]

**Figure S2c: Spread of event damage ratio (EDR, uncalibrated) and total damage ratio (TDR) per country in Oceania with Australia (OC). The plots are based on data from 11 countries. The EDR boxplots show the median (green line), the first and third quartiles (IQR, blue box), data points outside the IQR but not more than 1.5·IQR distance from either the first or the third quartile (black whiskers), and outliers (black circles). The additional markers show TDR before calibrated (green diamond) and after calibration (blue circle: RMSF optimized and red squares: TDR optimized).**

[Figure]

**Figure S2c: Spread of event damage ratio (EDR, uncalibrated) and total damage ratio (TDR) per country in the South Indian Ocean basin (SI). The plots are based on data from two countries. The EDR boxplots show the median (green line), the first and third quartiles (IQR, blue box), data points outside the IQR but not more than 1.5·IQR distance from either the first or the third quartile (black whiskers), and outliers (black circles). The additional markers show TDR before calibrated (green diamond) and after calibration (blue circle: RMSF optimized and red squares: TDR optimized).**

[Figure]

**Figure S2d: Spread of event damage ratio (EDR, uncalibrated) and total damage ratio (TDR) per country in the North West Pacific basin (WP). The plots are based on data from 11 countries. The EDR boxplots show the median (green line), the first and third quartiles (IQR, blue box), data points outside the IQR but not more than 1.5·IQR distance from either the first or the third quartile (black whiskers), and outliers (black circles). The additional markers show TDR before calibrated (green diamond) and after calibration (blue circle: RMSF optimized and red squares: TDR optimized).**
* * *
**3) ADDITIONAL MINOR CHANGES TO REVISED MANUSCRIPT**

3.1)
L.489: We realized that the sentence below was incomplete: The windfield by Holland (2008) does not completely neglect surface roughness, simulating 1-minute sustained wind speeds at 10 meter above "clear, flat terrain". What is neglected are variations in surface effects.
L.489: "The wind field model adapted from Holland (2008) does however not take into account **variation in** topography and surface roughness."

3.2)
L. 655 (Acknowledgements): "**We would like to thank Andrew Gettelman and one anonymous referee for their thorough and valuable reviews.**"

3.3)
In addition, some minor changes in spelling, grammar, punctuation, and figure numbers (due to new additional Figure 4) were introduced in the revised manuscript. These changes do not affect the meaning of the text in any way and are not listed here. However, all changes, including new references, are tracked in the revised manuscript attached.
* * *
**4) REVISED MANUSCRIPT (MARKED-UP VERSION)**

[revised manuscript text omitted]

**Main document changes and comments**

| Page 1: Deleted | Samuel Eberenz | 11/3/20 3:36:00 PM |
|---|---|---|

(TCFD, 2017)

| Page 1: Inserted | Eberenz et al. | 11/3/20 3:34:00 PM |
|---|---|---|

(Bloomberg et al., 2017)

| Page 2: Deleted | Eberenz et al. | 11/3/20 3:34:00 PM |
|---|---|---|

Natural risk

| Page 2: Inserted | Eberenz et al. | 11/3/20 3:34:00 PM |
|---|---|---|

Risk from natural hazards

| Page 2: Deleted | Eberenz et al. | 11/3/20 3:34:00 PM |
|---|---|---|

one

| Page 2: Inserted | Eberenz et al. | 11/3/20 3:34:00 PM |
|---|---|---|

it is inadequate to use a single

| Page 2: Deleted | Eberenz et al. | 11/3/20 3:34:00 PM |
|---|---|---|

might be inappropriate

| Page 2: Inserted | Eberenz et al. | 11/3/20 3:34:00 PM |
|---|---|---|

While the attribution of vulnerability to regional drivers is outside the scope of this study, the results can serve as a starting point for further research disentangling the socio-economic and physical drivers determining vulnerability to TC impacts locally and across the globe.

| Page 3: Inserted | Eberenz et al. | 11/3/20 3:34:00 PM |
|---|---|---|

Fig. 1a-d,

| Page 3: Inserted | Eberenz et al. | 11/3/20 3:34:00 PM |
|---|---|---|

Fig. 1e-f,

| Page 3: Inserted | Eberenz et al. | 11/3/20 3:34:00 PM |
|---|---|---|

Fig. 1g,

Fig. 1h,

[Figure]

[Figure]

| Page 3: Deleted | Eberenz et al. | 11/3/20 3:34:00 PM |
|---|---|---|

tracks, asset exposure, and a default impact function (a,

| Page 3: Moved to page 3 (Move #1) | Eberenz et al. | 11/3/20 3:34:00 PM |
|---|---|---|

Sect. 2.1 to 2.2.3

| Page 3: Deleted | Eberenz et al. | 11/3/20 3:34:00 PM |
|---|---|---|

). Resulting simulated damages are matched and compared to reported damage data from EM-DAT (b, 2.2.4 to 2.3.2). During calibration, impact modelling and damage comparison are repeated several times for regional impact functions with varied slope (2.3.3). The result is a set of best fitting impact functions for nine world regions (c). Finally, the calibrated impact functions are plugged into CLIMADA (d) to compute annual average damage per region.

| Page 3: Inserted | Eberenz et al. | 11/3/20 3:34:00 PM |
|---|---|---|

hazard (a), asset exposure (b), and a default impact function (c), c.f.

| Page 3: Moved from page 3 (Move #1) | Eberenz et al. | 11/3/20 3:34:00 PM |
|---|---|---|

Sect. 2.1 to 2.2.3

| Page 3: Inserted | Eberenz et al. | 11/3/20 3:34:00 PM |
|---|---|---|

. Resulting simulated damages (d) are compared to reported damage data from EM-DAT (e) for 473 matched TC events (f) by means of the damage ratio (g), c.f. Sect. 2.2.4 to 2.3.2. During calibration (h), steps (c) to (g) are repeated several times with varied impact functions for each region, optimizing the cost functions TDR and RMSF (c.f. Sect. 2.3.3) . The result is a set of best fitting impact functions for nine world regions (Sect. 3.2). Finally, the calibrated impact functions are plugged into CLIMADA once more (dashed arrow)to compute annual average damage per region (Sect. 3.3).

| Page 3: Deleted | Eberenz et al. | 11/3/20 3:34:00 PM |
|---|---|---|

does work

| Page 3: Inserted | Eberenz et al. | 11/3/20 3:34:00 PM |
|---|---|---|

works

| Page 3: Deleted | Eberenz et al. | 11/3/20 3:34:00 PM |
|---|---|---|

resolutions

| Page 3: Inserted | Eberenz et al. | 11/3/20 3:34:00 PM |
|---|---|---|

chosen resolution

| Page 3: Deleted | Eberenz et al. | 11/3/20 3:34:00 PM |
|---|---|---|

Simulated

| Page 3: Inserted | Eberenz et al. | 11/3/20 3:34:00 PM |
|---|---|---|

In the CLIMADA framework, damage is defined as the product of exposed assets and a damage ratio. The damage ratio is an impact function multiplied with hazard intensity.

In our case,

| Page 4: Deleted | Eberenz et al. | 11/3/20 3:34:00 PM |
|---|---|---|

computed

| Page 4: Inserted | Eberenz et al. | 11/3/20 3:34:00 PM |
|---|---|---|

simulated

| Page 4: Deleted | Eberenz et al. | 11/3/20 3:34:00 PM |
|---|---|---|

(1)

| Page 4: Inserted | Eberenz et al. | 11/3/20 3:34:00 PM |
|---|---|---|

damage is calculated as the product of total exposed asset values and

| Page 4: Inserted | Eberenz et al. | 11/3/20 3:34:00 PM |
|---|---|---|

. The mean damage ratio

| Page 4: Deleted | Eberenz et al. | 11/3/20 3:34:00 PM |
|---|---|---|

is determined by

| Page 4: Inserted | Eberenz et al. | 11/3/20 3:34:00 PM |
|---|---|---|

results from

| Page 4: Inserted | Eberenz et al. | 11/3/20 3:34:00 PM |
|---|---|---|

hazard intensity (

| Page 4: Deleted | Eberenz et al. | 11/3/20 3:34:00 PM |
|---|---|---|

 (hazard intensity

| Page 4: Deleted | Eberenz et al. | 11/3/20 3:34:00 PM |
|---|---|---|

an

| Page 4: Inserted | Eberenz et al. | 11/3/20 3:34:00 PM |
|---|---|---|

the

| Page 4: Deleted | Eberenz et al. | 11/3/20 3:34:00 PM |
|---|---|---|

(2) Absolute

| Page 4: Inserted | Eberenz et al. | 11/3/20 3:34:00 PM |
|---|---|---|

Finally,

| Page 4: Deleted | Eberenz et al. | 11/3/20 3:34:00 PM |
|---|---|---|

grid cell is computed by multiplying the mean damage ratio with the value of exposed assets at the grid cell. (3) The total damage per country and

| | | |
|---|---|---|
| **Page 4: Deleted** | **Eberenz et al.** | **11/3/20 3:34:00 PM** |

computed as the sum

| | | |
|---|---|---|
| **Page 4: Inserted** | **Eberenz et al.** | **11/3/20 3:34:00 PM** |

aggregated

| | | |
|---|---|---|
| **Page 5: Deleted** | **Eberenz et al.** | **11/3/20 3:34:00 PM** |

impact

| | | |
|---|---|---|
| **Page 5: Inserted** | **Eberenz et al.** | **11/3/20 3:34:00 PM** |

directly wind induced damage

| | | |
|---|---|---|
| **Page 5: Deleted** | **Eberenz et al.** | **11/3/20 3:34:00 PM** |

constraint

| | | |
|---|---|---|
| **Page 5: Inserted** | **Eberenz et al.** | **11/3/20 3:34:00 PM** |

constrained

| | | |
|---|---|---|
| **Page 5: Inserted** | **Eberenz et al.** | **11/3/20 3:34:00 PM** |

 sigmoidal

| | | |
|---|---|---|
| **Page 5: Deleted** | **Eberenz et al.** | **11/3/20 3:34:00 PM** |

We define a default impact function with $V_{thresh} = 25.7$ ms$^{-1}$ and $V_{half} = 74.7$ ms$^{-1}$ that is used for a first, uncalibrated, simulation of global TC damages, and as a starting point for calibration. While $V_{half}$ is fitted during the calibration process, we keep the lower threshold $V_{thresh}$ constant throughout the study.

| | | |
|---|---|---|
| **Page 5: Inserted** | **Eberenz et al.** | **11/3/20 3:34:00 PM** |

In a comparison of calibration results based on a sigmoidal impact function with a more complex 12-step staircase function, Lüthi (2019) found no improvement of calibration skill with the more complex function. Therefore, we use the sigmoidal function in this study. We define a default impact function with $V_{thresh} = 25.7$ ms$^{-1}$ and $V_{half} = 74.7$ ms$^{-1}$ that is used for a first, uncalibrated, simulation of global TC damages, and as a starting point for calibration. While $V_{half}$ is fitted during the calibration process, we keep the lower threshold $V_{thresh}$ constant throughout the study. This is based on the finding by Lüthi (2019) that the variation of more than one of the linearly dependent parameters most likely results in an overfitting during calibration, with physically implausible values for $V_{thresh}$ in some world regions.

| | | |
|---|---|---|
| **Page 6: Deleted** | **Eberenz et al.** | **11/3/20 3:34:00 PM** |

Damage

| Page 6: Inserted | Eberenz et al. | 11/3/20 3:34:00 PM |

Reported damage

| Page 6: Deleted | Eberenz et al. | 11/3/20 3:34:00 PM |

from natural catastrophes are records

| Page 6: Inserted | Eberenz et al. | 11/3/20 3:34:00 PM |

for disasters worldwide

| Page 6: Deleted | Eberenz et al. | 11/3/20 3:34:00 PM |

global

| Page 6: Deleted | Eberenz et al. | 11/3/20 3:34:00 PM |

on natural

| Page 6: Inserted | Eberenz et al. | 11/3/20 3:34:00 PM |

per event

| Page 6: Deleted | Eberenz et al. | 11/3/20 3:34:00 PM |

technological disasters

| Page 6: Inserted | Eberenz et al. | 11/3/20 3:34:00 PM |

country

| Page 6: Deleted | Eberenz et al. | 11/3/20 3:34:00 PM |

data at the country level

| Page 6: Inserted | Eberenz et al. | 11/3/20 3:34:00 PM |

estimates

| Page 7: Inserted | Eberenz et al. | 11/3/20 3:34:00 PM |

The distribution of EDR and TDR before calibration as well as TDR after calibration is shown per region in Figures 6 and S4 and per country in the supplementary Figure S2.

| Page 8: Inserted | Eberenz et al. | 11/3/20 3:34:00 PM |

(matched TC events)

| Page 9: Inserted | Eberenz et al. | 11/3/20 3:34:00 PM |

The distribution of uncalibrated EDR per region is shown in Figure 4.

[Figure]

**Figure 4: Spread of event damage ratio (EDR, boxplot) and total damage ratio (TDR) per region before calibration (V$_{half}$=74.7 ms$^{-1}$) per region. The plots are based on data from 473 TC events affecting 53 countries. The EDR boxplots show the median (green line), the first and third quartiles (IQR, blue box), data points outside the IQR but not more than 1.5·IQR distance from either the first or the third quartile (black whiskers), and outliers (black circles). The additional markers show TDR before calibrated (green diamond). The regions are the Caribbean with Central America and Mexico (NA1); the USA and Canada (NA2); North Indian Ocean (NI); Oceania with Australia (OC); South Indian Ocean (SI); South East Asia (WP1), the Philippines (WP2), China Mainland (WP3); rest of North West Pacific Ocean (WP4).**

RMSF (Fig. 5b) and

(Fig. A1)

and the cost functions RMSF

(Fig. 5c), indicating

show inter-regional differences with regard to

, show inter-regional differences.

 (Fig. 4 and 6).

| Page 9: Deleted | Eberenz et al. | 11/3/20 3:34:00 PM |
|---|---|---|

TDR is

| Page 9: Inserted | Eberenz et al. | 11/3/20 3:34:00 PM |
|---|---|---|

total simulated and normalized reported damage deviates

| Page 9: Deleted | Eberenz et al. | 11/3/20 3:34:00 PM |
|---|---|---|

different from one.

| Page 9: Inserted | Eberenz et al. | 11/3/20 3:34:00 PM |
|---|---|---|

(Table A2).

| Page 9: Deleted | Eberenz et al. | 11/3/20 3:34:00 PM |
|---|---|---|

in

| Page 10: Deleted | Eberenz et al. | 11/3/20 3:34:00 PM |
|---|---|---|

| Page 10: Inserted | Eberenz et al. | 11/3/20 3:34:00 PM |
|---|---|---|

| Page 10: Deleted | Eberenz et al. | 11/3/20 3:34:00 PM |
|---|---|---|

regions

| Page 10: Inserted | Eberenz et al. | 11/3/20 3:34:00 PM |
|---|---|---|

each region

| Page 10: Deleted | Eberenz et al. | 11/3/20 3:34:00 PM |
|---|---|---|

A1).

| Page 10: Inserted | Eberenz et al. | 11/3/20 3:34:00 PM |
|---|---|---|

4). There is no significant correlation between EDR and NRD (Fig. A3), suggesting that the over- and underestimation of simulated event damages is not related to TC severity.

| Page 10: Deleted | Eberenz et al. | 11/3/20 3:34:00 PM |
|---|---|---|

5c

| Page 10: Inserted | Eberenz et al. | 11/3/20 3:34:00 PM |
|---|---|---|

6c

| **Page 10: Deleted** | **Eberenz et al.** | **11/3/20 3:34:00 PM** |
|---|---|---|

| **Page 10: Inserted** | **Eberenz et al.** | **11/3/20 3:34:00 PM** |
|---|---|---|

| **Page 10: Deleted** | **Eberenz et al.** | **11/3/20 3:34:00 PM** |
|---|---|---|

5a

| **Page 10: Inserted** | **Eberenz et al.** | **11/3/20 3:34:00 PM** |
|---|---|---|

6a

| **Page 10: Deleted** | **Eberenz et al.** | **11/3/20 3:34:00 PM** |
|---|---|---|

5b

| **Page 10: Inserted** | **Eberenz et al.** | **11/3/20 3:34:00 PM** |
|---|---|---|

6b

| **Page 10: Deleted** | **Eberenz et al.** | **11/3/20 3:34:00 PM** |
|---|---|---|

Figure 4

| **Page 10: Inserted** | **Eberenz et al.** | **11/3/20 3:34:00 PM** |
|---|---|---|

Fig. 5

| **Page 10: Deleted** | **Eberenz et al.** | **11/3/20 3:34:00 PM** |
|---|---|---|

Plots

| **Page 10: Inserted** | **Eberenz et al.** | **11/3/20 3:34:00 PM** |
|---|---|---|

The sensitivity

| **Page 10: Deleted** | **Eberenz et al.** | **11/3/20 3:34:00 PM** |
|---|---|---|

as functions of

| **Page 10: Inserted** | **Eberenz et al.** | **11/3/20 3:34:00 PM** |
|---|---|---|

to changes in

| **Page 10: Deleted** | **Eberenz et al.** | **11/3/20 3:34:00 PM** |
|---|---|---|

are provided

| Page 10: Inserted | Eberenz et al. | 11/3/20 3:34:00 PM |
|---|---|---|

is visualized

[revised manuscript text omitted]

6c).
* * *
**Page 17: Inserted**    Eberenz et al.    11/3/20 3:34:00 PM

7c) that were not as exposed to surge as Leyte province.
* * *
**Page 17: Inserted**    Eberenz et al.    11/3/20 3:34:00 PM

The flood damage caused by Ketsana is a showcase example for severe damages associated with a TC with relatively low wind speeds, that is, an event that cannot be adequately reproduced with a wind-based impact function.
* * *
**Page 17: Deleted**    Eberenz et al.    11/3/20 3:34:00 PM

(Espada, 2018; IFRC, 2009).
* * *
**Page 17: Inserted**    Eberenz et al.    11/3/20 3:34:00 PM

(Espada, 2018; IFRC, 2009; Yumul et al., 2012).
* * *
**Page 17: Inserted**    Eberenz et al.    11/3/20 3:34:00 PM

variation in
* * *
**Page 18: Inserted**    Eberenz et al.    11/3/20 3:34:00 PM

4,
* * *
**Page 19: Inserted**    Eberenz et al.    11/3/20 3:34:00 PM

[revised manuscript text omitted]

---

## Author Response (AR2)

**AUTHORS' RESPONSE**

**Research article**: "Regional tropical cyclone impact functions for globally consistent risk assessments" (Nat. Hazards Earth Syst. Sci. Discuss., https://doi.org/10.5194/nhess-2020-229; in review, submitted on 09 July 2020)
**Authors:** Samuel Eberenz, Samuel Lüthi, David N. Bresch

We thank both referees for their second round of review, and the comments by the first referee which have further improved the spelling and readability of the manuscript. These comments are listed below directly followed by our responses in *blue and italic* and changes to the manuscript in **blue and bold**. The marked-up version of the revised manuscript attached below shows all proposed changes.
* * *
**TABLE OF CONTENT**
* * *
**1) RESPONSE TO COMMENTS BY ANONYMOUS REFEREE #1 (2nd iteration: minor revisions)**

1.1)     para 140, line 2 - I think 'disaster' should be hazard?

*We agree to this comment and will correct the sentence as follows:*
L. 137f:
"Asset exposure for the assessment of direct economic risk is represented by the spatially explicit monetary value potentially impacted by a  **hazard**"

1.2)     Data and methods - I think that some consistency is needed in the data and methods section as well as the conclusion in the use of passive vs active voice. In some sections it is written passively (para 85) and in other paragraphs (175 and 180) 'we' is used extensively. I do not have a problem with the active voice as it makes one realise that there are people behind the science, ha!, but I think it could be minimised when talking

about objective data treatments vs the choices made in the study or results that you uncover. This could be very easily and quickly revised and it will just help the narrative flow.

*Thank you for pointing out inconsistencies in the use of active and passive voice. We have revised the manuscript, replacing most uses of "we" with the passive voice in the Section **Data and methods** as well as the first sentences of the **Conclusion**, where the method is summarized. In some places of the method section as well as the case study and the main part of the conclusion, we still find the active voice most adequate and leave these parts unchanged. Please find all changes in the text below. Please also note that some sentences required some paraphrasing to maintain or even improve the readability of the sentences in the passive voice:*

***Data and methods:***
L. 104f: " **Here,** CLIMADA **was used** for the pre-processing […]"

L. 123f: "**Wind** speed **was simulated** at a horizontal resolution of 10 x 10 km from historical TC tracks as a function of time, […]"

L. 129f: "[…]  4'098 historical TC tracks from 1980 to 2017 **were selected** based on data completeness criteria […]"

L. 132: "Out of the 4'098 TCs,  **a total number of** 1'538 landfalling events with the potential of causing damage **was identified**."

L. 171ff: "Therefore,  **a** sigmoidal function **is applied** in this study. **The** default impact function with $V_{thresh}$ = 25.7 ms$^{-1}$ and $V_{half}$ = 74.7 ms$^{-1}$  is used for a first, uncalibrated, simulation of global TC damages, and as a starting point for calibration. While $V_{half}$ is fitted during the calibration process,  the lower threshold $V_{thresh}$ **is kept** constant throughout the study."

L. 178ff: " **On the chosen 10 km by 10 km grid, single buildings are not resolved. Therefore, damage is aggregated over several buildings in a grid cell and not all buildings are expected to be damaged to the same degree.** However, the wind-speed dependent impact function is also implicitly accounting for the damage caused by storm surge and torrential rain, when calibrated against reported damage data. For these two reasons, we allow for values of $V_{half}$ lower and larger than the literature range for pure wind induced building damage in the calibration. **To find** the functional slope best fit to simulate the direct economic damage of TCs in  **a region,** $V_{half}$ **is varied step-wise with** $V_{half} > V_{thresh}$ (**c.f.** Sect. 2.3.3**, Calibration of regional impact functions**)."

L. 203f: "Due to a lack of global time series of wealth data,  **reported damage is normalized by means of** a GDP scaling "

L. 226f: "Generally,  the difference between simulated and reported damage per matched event  span several orders of magnitude."

*Conclusion and outlook:*
L. 613ff: "In this article,  **the** global  assessment **of TC risk was improved** by regionalizing the vulnerability component of the TC impact assessment. To better account for regional differences,  a TC impact model **was calibrated** by fitting regional impact functions."

1.3)   para 275 am measure - a measure

L. 276: "The RMSF is  measure of the […]"

1.4)   para 505 that [space] sector

L. 505: *We added the missing space between the words* "that" *and* "sector".
* * *
2) ADDITIONAL PROPOSED MINOR CORRECTIONS TO REVISED MANUSCRIPT
(2nd iteration: minor revisions)

L.134: *add space:*
" **world map**"

L. 532: *correct typo:*
"a**n** overestimation"

L. 650, 656, 662: *correct spelling of "color" in figure captions:*
"colo**u**r"

L. 696: *add missing space:*
"[…] largest normalized reported  **damage: storm** name […]"

L. 716f:
*In report #2, both referees chose the following option:* "Anonymous in Acknowledgements of published article: YES"
*We therefore adjust the acknowledgements as follows:*

[revised manuscript text omitted]